# Different Factors Influencing Postural Stability during Transcutaneous Electrical Stimulation of the Cervical Spinal Cord

**DOI:** 10.3390/jfmk9030142

**Published:** 2024-08-22

**Authors:** Leisan Bikchentaeva, Margarita Nikulina, Anna Shulman, Maxim Baltin, Angelina Zheltukhina, Elena Semenova, Viktoriya Smirnova, Svetlana Klepikova, Tatyana Baltina

**Affiliations:** 1Institute of Fundamental Medicine and Biology, Kazan Federal University, Karla Marksa 76, 420015 Kazan, Russia; leysanbm@mail.ru (L.B.); margo.nikulina.02@bk.ru (M.N.); ani_07@mail.ru (A.S.); angelina7385@yandex.ru (A.Z.);; 2Sport Science Department, Sirius University of Science and Technology, Olympic Ave. 1, 354349 Sirius Federal Territory, Russia; baltin.me@talantiuspeh.ru; 3Lobachevskii Institute of Mathematics and Mechanics, Kazan Federal University, Kremlevskaya 35, 420008 Kazan, Russia; elena.semionova2011@gmail.com (E.S.); yaikovavictoriya@mail.ru (V.S.)

**Keywords:** postural stability, transcutaneous electrical stimulation, cervical spinal cord, spectral analysis, healthy participants

## Abstract

Transcutaneous spinal cord stimulation (tSCS) is a promising noninvasive alternative to epidural stimulation. However, further studies are needed to clarify how tSCS affects postural control. The aim of this study was to investigate the effect of transcutaneous cervical spinal cord stimulation on postural stability in healthy participants via computerized stabilization. The center of pressure and the frequency spectrum of the statokinesiogram were assessed in 14 healthy volunteers under tSCS conditions with frequencies of 5 Hz or 30 Hz, subthreshold or suprathreshold stimulus strength, open or closed eyes, and hard or soft surfaces in various combinations. The results revealed that not all the changes in the center of the pressure oscillations reached statistical significance when the tSCS was used. However, tSCS at a frequency of 30 Hz with a suprathreshold stimulus strength improved postural stability. The use of subthreshold or suprathreshold tSCS at 5 Hz led to a shift of 60% of the signal power to the low-frequency range, indicating activation of the vestibular system. With tSCS at 30 Hz, the vestibular component remained dominant, but a decrease in the proportion of high-frequency oscillations was observed, which is associated with muscle proprioception. Thus, transcutaneous electrical stimulation of the cervical spinal cord may be an effective method for activating spinal cord neural networks capable of modulating postural control.

## 1. Introduction

Balance control is a complex process that involves the voluntary and involuntary maintenance of an upright posture, both in the absence of movement and during movement. This also requires the ability to respond effectively to external and internal signals that can destabilize posture [1]. The visual, vestibular, and proprioceptive systems provide information that integrates into the somatosensory cortex to control muscles, posture, and appropriate motor actions [2]. Disruption of any of the information transmission points can lead to disruption of balance function.

The cervical spine plays an important role in the proper functioning of the vestibular and proprioceptive systems, which in turn affects balance, coordination of movement and cenesthesia [3,4,5]. The cervical spinal cord has been shown to contribute to the fine control of movement because it contains many proprioceptors [6]. Cervical proprioception is particularly important for spatial orientation because it provides a frame of reference for how the head is positioned and moves relative to the trunk. While vestibular afferents encode active and passive head movements, cervical proprioception acts as a reference point for perceiving the actual position of the head relative to the trunk. This function is essential for the correct interpretation of visual and vestibular inputs, such as gaze stabilization during head movements [7]. Proprioception is also important for the execution of coordinated, corrective, and targeted intersegmental movements [8]. The neuromuscular and sensory pathways of the cervical spinal cord and brain form connections that are important components for maintaining postural tone and neuromuscular coordination [8,9]. The presence of such connections allows the use of transcutaneous spinal cord stimulation (tSCS) to modulate descending pathways. In neurologically healthy participants, tonic activation of spinal cord networks through multiple sites of tSCS has been shown to facilitate both the spinal reflex and corticospinal pathways [10].

The tSCS method improves motor function in neurological patients [11,12,13,14], facilitates pacing movements in healthy participants [15], and contributes to the improvement in balance function [16,17]. tSCS, like epidural stimulation, is capable of altering excitability through the recruitment of posterior-root fibers to activate interneuronal networks, potentiate the generation of postsynaptic excitatory potentials, and contribute to the excitation threshold of interneurons and motor neurons. This allows the spinal neuronal network to respond to the descending impulse and increases the overall excitability of the spinal network and potentially the motor cortex [18,19]. However, unlike epidural stimulation, tSCS does not require surgical implantation of electrodes. A special configuration of pulses with carrier frequencies from 5 to 10 kHz minimizes discomfort when the tSCS is used [15,20,21]. The noninvasiveness and painlessness of the method allows it to be used on both healthy participants and patients with SCI. Spinal stimulation of lumbosacral enlargement effectively promoted movement in participants whose legs were in a neutral gravity position. Furthermore, the combination of sensory and spinal cord stimulation has been shown to elicit synergistic effects [22]. The tSCS has also been found to be useful by restoring the functions of the upper extremities [23,24,25]. tSCS appears to modulate the cervical–lumbar connection and promotes improvements in motor function [26]. Noninvasive stimulation at multiple spinal sites at the C5–C6, Th11–Th12, and L1–L2 levels improves walking quality in healthy participants by increasing the amplitude of involuntary stepping movements [27,28]. Even in the absence of supraspinal control, tSCS modulates spinal networks and improves postural control [16]. The effect of tSCS may depend on the frequency applied: 5 Hz tSCS has been shown to produce a muscle response. The application of higher frequencies results in a more stable position of the center of pressure (CoP) and makes it easier for patients with SCI to stand independently [16].

The positive effects of tSCS have been shown in many studies, but the mechanisms of postural stability improvement still require further investigation. To understand the mechanisms of postural control, it is necessary to conduct studies on healthy participants. We hypothesized that percutaneous electrical stimulation of the cervical spinal cord would be an effective way to activate neural–spinal locomotor networks to modulate the postural control system. Furthermore, the contribution of information from different sensory systems may change under the influence of tSCS, which influences the mechanisms of postural stability maintenance.

The aim of this study was to investigate the effects of transcutaneous electrical stimulation of the cervical spinal cord at different frequencies and strengths on the mechanisms of postural stability in healthy participants.

## 2. Materials and Methods

### 2.1. Participants of This Study

A total of 14 volunteers (3 males, 11 females, aged 19–27 years) were included in this study. The height of the participants was 163.7 ± 10.7 cm, and their body mass was 59.2 ± 16.9 kg. All the participants rated themselves as healthy on the day of this study. The specific inclusion criteria were the absence of implanted metallic or electrical devices, pregnancy, and musculoskeletal or neurological disorders that could affect postural balance and well-being on the day of this study. The anthropological data of all the participants are also presented in Appendix A.

This study was conducted with the informed voluntary consent of the participants in accordance with the Declaration of Helsinki developed by the World Medical Association. This study protocol was approved by the Local Ethical Committee of Kazan Federal University (protocol No. 34 of 27.01.2022).

### 2.2. Transcutaneous Electrical Stimulation of the Spinal Cord

In this study, a five-channel BIOSTIM-5 stimulator (Cosyma Ltd., Moscow, Russia) was employed. A stimulating cutaneous round electrode (cathode) 32 mm in diameter with an adhesive layer (PG479/32, Fiab, Mitcham, UK) was positioned on the skin between the spinous processes of the C5 and C6 vertebrae. Rectangular electrodes (anodes) 45 × 80 mm in size with an adhesive layer (PG472W, Fiab, UK) were placed symmetrically on the clavicles (Figure 1a) [29].

The stimulation was conducted via rectangular bipolar pulses with a duration of 1 ms, filled with a carrier frequency of 10 kHz, as previously described [30]. The current intensity was selected for each participant by gradually increasing the current until a motor response was observed in all the muscles of the upper limbs that were being studied: the m. flexor carpi ulnaris and the m. extensor carpi radialis, bilaterally, without allowing the appearance of any unpleasant sensations. The current intensity was subsequently reduced by approximately 10% (subthreshold stimulation) or increased by approximately 50% (threshold stimulation). The current intensity varied from 12 to 27 mA. The stimulation frequency was set at 5 Hz or 30 Hz, depending on the experimental conditions.

The electromyographic (EMG) activity was recorded and subsequently processed via a Neuro-MVP-8 electroneuromyograph (Neurosoft, Ivanovo, Russia), which is an 8-channel amplifier with Neuro-MVP software (v. 4.2). The EMG data were filtered via a 50 Hz notch filter and a 20–1000 Hz bandpass filter. The data were recorded at a frequency of 4 kHz, exported, and analyzed with Neuro-MVP software. Surface electrodes (BE-1, NS990998.028, Neurosoft, Moscow, Russia) were placed on the bellies of the right and left muscles: m. flexor carpi ulnaris and m. extensor carpi radialis; the interelectrode distance was 20 mm (Figure 1b,c), and the placement site was preliminarily determined by palpation. After placing the electrodes, the quality of their installation was checked by measuring the subelectrode impedance. Muscle responses were recorded without changing the position of the EMG electrodes [31,32]. The threshold stimulation intensity for the occurrence of motor responses was assessed twice. The first test was conducted with 5 Hz stimulation, and the second test was conducted with 30 Hz stimulation on a different day.

### 2.3. Stabilography

For stabilometry, the Stabilan-01-2 force platform system (Ritm LLC., Taganrog, Russia) with StabMed 2.13 software was used [33]. The system recorded the position of the center of pressure (CoP) with a sampling frequency of 50 Hz and a resolution of <0.01 mm. For clarity, Figure 2 shows an example of a stabilogram and a statokinesiogram recording.

On the stabilographic platform, on which a person is in a standing position, force sensors that measure the reaction of the person’s foot support are attached. The two-coordinate computer platform ensures the recording and analysis of the trajectory of the center of pressure (CoP) exerted by a person on the horizontal working surface of the force-coordinate platform. The registration CoP fluctuations and all calculations were provided by the corresponding software of the StabMed 2 stability analyzer.

When the statokinesiogram was processed, the following static parameters were analyzed: the area of the ellipse, an indicator characterizing 90% of the surface occupied by the statokinesiogram and reflecting the area of the subject’s support during the examination (ELLS, mm^2^); the root-mean-square deviation of the CoP along the frontal (X) and sagittal (Y) axes (Qx and Qy, mm); and the dynamic parameters, the average linear velocity of the CoP movement (ALV, mm/s) and the average angular velocity of the CoP movement (AAV, deg/s) (the formulas for the calculation are provided in Appendix A).

### 2.4. Spectral Analysis of the Stabilogram

Amplitude spectra of the fast Fourier transform with a Hamming window were found for signals from the force platform via a script developed in MATLAB R2019a [34] (Figure 3). The frequency was plotted on the abscissa axis, and the region of interest was defined as 0.02–5 Hz. The signal amplitude was plotted on the vertical (ordinate) axis. The frequency corresponding to 60% of the power was identified at the point corresponding to 60% of the signal area across the entire frequency range of 0.02–5 Hz. This value represented 60% of the power of the stabilogram spectrum, which was designated 60%Pw. The power spectrum was divided into four frequency ranges: the ultralow-frequency ranges from 0.02 to 0.2 Hz (Pw1), the low-frequency range from 0.1 to 1.0 Hz (Pw2), the mid-frequency range from 0.5 Hz to 2 Hz (Pw3), and the high-frequency range from 2 Hz to 5 Hz (Pw4).

The ultralow-frequency range is associated with the contribution of visual information to the oscillations of the center of pressure, the low-frequency range with vestibular information, the mid-frequency range with somatosensory and cerebellar information, and the high-frequency range with proprioceptive information [35,36]. For each zone, the maximum amplitude was identified. The data for the maximum amplitudes of all participants across all ranges were presented via a violin plot (Figure 4).

Five violin plots were generated before, during, and after stimulation. Additionally, the data collected during stimulation were divided into three equal one-minute intervals. Accordingly, all five signals, each with a duration of one minute, were processed in accordance with the aforementioned methodology. The violin plot enabled the evaluation of two key aspects of the data: the kernel density, which is represented by the ‘violin’ shape, and the data density at a specific point, which is reflected in the ‘violin’ width at that point.

### 2.5. The Experimental Protocol and Data Collection

(1) The motor response thresholds of the right and left forearm muscles, specifically the m. flexor carpi ulnaris and m. extensor carpi radialis, were elicited among the participants to determine the optimal stimulation intensity. Further details regarding this procedure can be found in Section 2.2.

(2) The participants were asked to stand on the force platform in a neutral position, with their arms at their sides and their feet in a heel-together, toes-apart position (commonly referred to as the “European position”, with heels positioned 2 cm apart and toes at an angle of 30°), aligning the sagittal plane with the anteroposterior axis of the force plate (Figure 5a,b). Centering was performed prior to testing to ensure that the person’s center of pressure was aligned with the origin.

(3) A stabilography test was conducted under the following conditions for all participants: (1) a 1-minute trial without stimulation; (2) for 3 minutes with stimulation; and (3) for 1 minute without stimulation immediately after stimulation. In each session, we conducted a total of four tests (Figure 6).

These methods were as follows: standing with eyes open (Figure 6a); standing with eyes closed (Figure 6b); standing on a soft, unstable support (foam pad 49 cm L × 49 cm W × 18 cm H with foot position marks) with eyes open (Figure 6c); and standing with eyes closed (Figure 6d). Each session lasted a maximum of 90 minutes, including setup, calibration, and testing. We took five-minute breaks between recordings to minimize fatigue. The participants were allowed to walk or rest on the force platform between recordings.

We conducted this study under the following conditions: eyes open (EO)/eyes closed (EC) and hard surface (HS)/soft surface (SS) in different combinations: EOHS; ECHS; EOSS; and ECSS. The order of the four recordings was randomized. After a short break (≤5 min), another random order of these four recordings followed. A total of two sessions with subthreshold and suprathreshold stimulus strengths were performed in one day, with 15 minutes in between. Each set of these four conditions was considered an independent series as a test. Stimulation at 5 Hz or 30 Hz was performed on different days.

Thus, for each participant in all experimental conditions, we obtained four tests with the following tSCS parameters: 5 Hz with subthreshold (*n* = 11) and suprathreshold (*n* = 11) stimulation and 30 Hz with subthreshold (*n* = 14) and suprathreshold (*n* = 14) stimulation.

### 2.6. Statistical Analysis

Statistical analysis of the data was performed via the MedStat and MedCalc v.15.1 software packages (MedCalc Software bvba, Ostend, Belgium). Nonparametric indicators of the results of stabilometry were evaluated and are presented in the form of medians and interquartile ranges, as well as in the tables in Appendix A in the form of means (M) and standard deviations (±SD). Violin plots were used for amplitude spectrum result visualization. Data processing and visualization of the amplitude spectrum results were performed in MATLAB R2019a. A nonparametric approach, specifically the Wilcoxon test, was employed for the purpose of comparison between the various experimental conditions. The significance of the differences between the parameters during stimulation with varying strengths and frequencies was calculated via the Mann–Whitney U test. The threshold level of statistical significance was accepted when the criterion value was *p* < 0.05.

## 3. Results

### 3.1. Use of 5 Hz

While some changes were statistically significant, this was not the case for all changes when 5 Hz transcutaneous electrical spinal cord stimulation (tSCS) was used at subthreshold and suprathreshold stimulus strengths (see Appendix A). Clearly, 5 Hz tSCS improved postural stability in the eyes-closed soft surface (ECSS) condition. Most changes in the static and dynamic parameters of the center of pressure (CoP) oscillation demonstrated this improvement. For example, subthreshold tSCS stimulation decreased the average linear velocity (ALV) (Figure 7).

The figure shows that the ALV increased more than three-fold (*p* < 0.05) under the challenging test conditions (hard/soft surface). However, with subthreshold tSCS at 2 min of stimulation, the ALV decreased by 13% (*p* < 0.05). Additionally, the average angular velocity (AAV) decreased by 33% under ECSS conditions (*p* < 0.05) and increased by 4% under stimulation (*p* = 0.36), although this difference was not statistically significant.

At 5 Hz, the tSCS of the suprathreshold intensity resulted in a clear shift toward stability under the conditions of ECSS. This was evidenced by an increase in ELLS, Qx, and Qy, which was not statistically significant (Figure 8a). Additionally, the dynamic parameters demonstrated a notable decrease in the ALV by 12% (*p* < 0.05) and an increase in the AAV by 10% (*p* > 0.05). This was a significant change compared with the values at subthreshold stimulus strength (Figure 8b). However, it should be noted that at 5 Hz tSCS, regardless of the stimulation intensity, an increase in the root-mean-square deviation in the frontal plane with open eyes was observed (Figure 9, Appendix A).

Overall, with a subthreshold of 5 Hz tSCS, 60% of the spectral power (60% Pw) was 0.4 [0.3; 0.5] Hz in the sagittal plane and 0.3 [0.3; 0.5] Hz in the frontal plane, corresponding to the low-frequency range (Pw2) that characterizes the predominant influence of the vestibular system. With eyes closed, the 5 Hz subthreshold tSCS resulted in a shift of 60% Pw toward the mid-frequencies (Pw3): 0.5 [0.4; 0.5] Hz and 0.4 [0.4; 0.5] Hz in the sagittal and frontal planes, respectively (Figure 10). This shift was due to an increase in the proportion of Pw3 to 32.7 [29.5; 36.2] % and Pw4 (high frequencies) to 9 [8.1; 9.3] % in the sagittal plane; in the frontal plane, the proportion of Pw3 increased to 31 [25.6; 34.7] %, Pw4 to 7.2 [6.2; 8.9] % (Appendix A).

At the suprathreshold tSCS intensity, the 60% Pw range shifted to 0.4 [0.4; 0.5] Hz in the sagittal plane and 0.5 [0.4; 0.5] Hz in the frontal plane in the eyes-closed test (Figure 11). This shift was attributable to an increase in the proportion of Pw4(S) to 7.9% [7.9; 8.2%] (Figure 11, Appendix A). In general, during suprathreshold tSCS, the vestibular component prevailed in the spectrum, and the muscular component was added.

### 3.2. Use of 30 Hz

The results of the tSCS at subthreshold and suprathreshold intensities of 30 Hz revealed that not all the obtained changes were statistically significant (see Appendix A). tSCS at the subthreshold intensity resulted in statistically significant changes in the tests with eyes open. In particular, on a hard surface, the ELLS decreased by 40% (*p* < 0.05), and the Qy and Qx decreased by an average of 23% (*p* < 0.05). On a soft surface, the ELLS decreased by 9%, and the ALV decreased by 8% (*p* < 0.05) (Appendix A). With the eyes closed, subthreshold stimulation at 30 Hz resulted in a decrease in ALV in the hard (by 14%) and soft (by 5%) surface tests, as well as an increase in ELLS (by 37%), Qy and Qx (11% and 20%, respectively). However, these changes were not statistically significant (Figure 12).

At 30 Hz, tSCS with the suprathreshold intensity was shown to improve postural stability, as evidenced by changes in the static and dynamic parameters of CoP oscillations (Appendix A). Significant changes were obtained in the ECHS test. Thus, at the second minute of stimulation, the ELLS decreased by 35% (*p* < 0.05), the Qy decreased by 27%, the Qx decreased by 14%, and the ALV decreased by 18% (*p* < 0.05) (Figure 13, white bars with stripes). Under ECSS conditions, ELLS decreased by 10%, Qy increased by 2%, Qx decreased by 8%, the average linear velocity decreased by 6%, the average angular velocity increased by 2%, and the changes were statistically insignificant (Figure 13, gray bars with stripes).

During suprathreshold tSCS intensity, compared with subthreshold tSCS intensity, a statistically significant decrease in Qy was obtained (Figure 14). Under ECHS conditions, the decrease in Qy was two-fold greater at the suprathreshold stimulus intensity (*p* < 0.05).

The analysis of the results of the change in the stabilogram spectrum demonstrated that with tSCS at the subthreshold stimulus strength, the 60% Pw range decreased toward lower frequencies in the eyes-closed test in the sagittal and frontal planes. The 60%Pw(S) was 0.4 [0.4; 0.6] Hz, and the 60%Pw(F) was 0.4 [0.3; 0.5] Hz (Figure 15). This change resulted from a decrease in the proportion of Pw3(S) to 32.9 [27.9; 38.5]% and a decrease in Pw3(F) to 28.4 [24; 38.3]%. These results are presented in Appendix A.

At a suprathreshold 30 Hz TSCS stimulus strength during the eyes-closed test on a soft surface, 60% Pw decreased toward lower frequencies, measuring 0.5 [0.5; 0.7] Hz in the sagittal plane and 0.5 [0.5; 0.7] Hz in the frontal plane. This shift occurred due to a decrease in the proportion of Pw4(S) to 8.9 [7.2; 11.3]% and a decrease in the proportion of Pw4(F) to 7.8 [6.5; 10.2] % (Figure 16, Appendix A).

In general, with 30 Hz stimulation, the ratio of spectral powers before and during stimulation was maintained. The proprioceptive component increased in the sagittal and frontal planes (Appendix A).

## 4. Discussion

With the widespread use of spinal cord electrical stimulation in locomotion and postural control studies, much of the role of spinal networks remains unclear. Transcutaneous spinal cord stimulation (tSCS) offers a significant advantage: this noninvasive method activates spinal networks, including those in healthy individuals, enabling the study of spinal cord neural networks in humans. The purpose of this study was to investigate the effects of cervical spinal cord tSCS on the mechanisms of maintaining postural stability in healthy participants. We assessed the postural control of healthy participants via static and dynamic stabilographic parameters and interpreted our results in terms of changes in afferent input activity during simple and complex trials. Our results demonstrated that cervical spinal cord tSCS altered the feedback of postural control, resulting in slower (lower mean CoP displacement rate, ALV), more stable (lower mean CoP standard deviation, Q), and more regulated (lower CoP deviation, especially in the frontal plane, and reduced ellipse) CoP oscillations in almost all tests, indicating an overall positive effect on postural stability. However, we can identify some peculiarities in the effects of the cervical tSCS at various frequencies and strengths.

### 4.1. Postural Stability of Healthy Participants

In studies of human posture control, it is assumed that information transmitted by individual receptors is less relevant than information transmitted through large groups of receptors distributed throughout the body and then integrated by the CNS [37,38]. The integration of multiple inputs from the vestibular, visual, and somatosensory systems that provide information about the CoP position, velocity, and acceleration of CoP displacement can enable excellent body stabilization by reducing CoP oscillations. The existence of a single posture control center capable of integrating information from sensory inputs and producing appropriate motor commands, which are apparently optimal when the center receives as much information as possible, has been suggested [39]. Precise neural control can be judged by almost imperceptible body sway when the body is upright on solid support in healthy people and when the input signals from various receptors change [40,41]. Standing on a soft surface results in more body sway and apparent instability as the difficulty of the task increases and the muscles of the limb become involved in maintaining balance [42,43,44].

Our data confirmed that with open eyes on a hard surface, the time of quiet standing without additional restrictions on postural stability is greatest. With closed eyes on a soft surface and their combination, static and dynamic indices show serious changes. However, even in this case, all our participants had a CoP deviation within the normal range, demonstrating healthy postural control. To analyze different experimental conditions, we evaluated the changes in postural control related to vision, the support surface, and the cervical spinal cord tSCS to identify any potential relevance for clinical practice.

### 4.2. Cervical Spinal Cord and Postural Stability

The sensorimotor control of sustained upright posture as well as locomotion is highly dependent on cervical afferent information, which plays a crucial role in three reflexes that influence head position, eye position, and postural stability: the cervicocolic reflex, the cervicocular reflex, and the tonic cervical reflex [3]. These reflexes integrate with the vestibular and visual systems with CNS involvement to maintain proper postural control [7,45,46]. Cervical–lumbar connections and connections between the cervical and thoracic spinal cord form an important component of maintaining postural tone and neuromuscular coordination. The neuromotor control of the muscles of the lower extremities and spinal cord is coordinated with the postural reflexes of the head and neck through cerebellar integration. In this way, a comprehensive network of involuntary postural control develops [6].

As mentioned above, considering that postural control is not a simple summation of static reflexes but rather a complex skill based on the interaction of dynamic sensorimotor processes [39], we suggest that additional activation of the cervical spinal cord by tSCS, which changes the activity of both vestibular input and descending propriospinal tracts, may have a positive effect on postural control. The results revealed that the cervical spinal cord is activated by both rhythmic arm rotation [47,48] and tonic tSCS, which modulates the activity of lumbar networks [49]. This connection improves walking function after neurological impairment [14,50]. These findings suggest that engaging cervical spinal cord networks through tSCS may not only improve cervical–lumbar connectivity but also improve the ability to stand and walk [26]. There are positive examples of such effects in pathological conditions, such as improved neural transmission across lesions after SCI to restore motor control in the upper extremities [25,51]. In early studies, we reported a downward facilitating effect with active involvement of the upper extremities when stimulation was applied. Combining tSCS with the Jendrassik technique (a participant interlocks the fingers of two hands in a "lock" in front of the chest and then attempts to break the lock) produced a significant facilitatory effect on the responses of the lower extremity muscles in a subject diagnosed with complete paralysis (AIS-A) [52]. The majority of studies where tSCS is used involve the thoracic and lumbar spinal cord. They are mostly related to the initiation of motor activity of the lower limbs for walking [11,16,17]. tSCS results in the recovery of an upright posture in patients with motor impairments. In healthy participants, it reduces postural stability, as suggested by an additional supraspinal effect on antigravity stability mechanisms, which are impaired in patients [53]. However, tSCS can also modulate the integration of voluntary top-down commands and sensory cues and alter muscle activation during a postural task, even in healthy participants [54].

Interestingly, in our study, when stimulating the cervical spinal cord, we observed an accumulative positive effect on postural stability if it was possible to describe it in this way. According to the data in Appendix A, our participants’ CoP oscillations decreased from trial to trial. For example, the area of the ellipse decreased by almost half. We compare the ELLS values under 5 Hz stimulation conditions with the subthreshold force in the EOHS trial and at 30 Hz with the suprathreshold force in the EOHS trial. We think that several interpretations are possible here. Either it is an effect of stimulation, training rather than stimulation, or a combination of both. However, since we did not perform a control without stimulation while maintaining a common protocol, which is a limitation of this study, we cannot make an accurate assessment. It has been shown that as participants become increasingly accustomed to changing stance conditions, responses include reduced muscle activation and improved posture control [55]. These changes may be the result of optimizing the motor system to maintain balance, reduce energy expenditure, and meet task demands [56,57]. Nevertheless, we observed an improvement in postural stability from the first trial to the last trial. Figure 9 shows that, for example, in open-eye trials with 5 Hz stimulation, the root-mean-square deviation in the frontal plane increased, which was associated with an increase in the contribution of the thigh or trunk muscles [58]. The average linear velocity (ALV) also changed and increased when the eyes were closed under these conditions (Figure 7). It can be assumed that at the beginning of the experiment, in participants with cervical tSCS, the change in ALV and the increase in the spread of frontal and sagittal oscillations reflect the tension of postural regulatory mechanisms [37]. The increased complexity of the task in the soft surface trial led to an initial Qy of more than 10 mm, which exceeds the norm [58]. However, further complications did not lead to such a result. Conversely, from trial to trial, we observed improvements in postural stability in static and dynamic indices during tSCS, and the best effect was observed in the most “unstable” conditions: the 30 Hz suprathreshold stimulation. In addition, during TSCS, we observed a higher percentage of the CoP signal in the low-frequency (0.2–1 Hz) range and a lower percentage in the high-frequency (2–5 Hz) range. Low- and moderate-frequency ranges can be associated with the vestibular and muscular proprioceptive systems, respectively [59]. The activation of vestibular reflexes is critically important, especially with closed eyes and on a soft surface, since visual and proprioceptive signals are absent, and there is additional activation of the cervical spine [60]. Our studies have shown that the vestibular system is the most active under such conditions. The vestibular system is believed to function as an internal cue, possibly suppressing misleading additional cues when present [61].

### 4.3. The tSC at 5 Hz and 30 Hz Frequencies

Many experts assume that after current strength, frequency is the most important parameter for achieving effective stimulation, although there are no published studies supporting this assertion. It is assumed that higher frequencies during epidural spinal cord stimulation in the lumbar region alter the central state of spinal circuits [62]; stimulation at frequencies of 5–15 Hz elicits sustained tonic responses. This facilitates tonic extensor activity specific to postural control, whereas stimulation at frequencies of 25–50 Hz induces rhythmically alternating flexion/extension activity [63]. In the majority of studies, it is generally accepted that the stimulation frequency required to enhance the motor response is 30 Hz. This frequency is effective in restoring motor function in patients with spinal cord injury during epidural stimulation [64,65,66], during tSCS [11,26] and in healthy participants [53,54,67]. A more pronounced muscle response on EMGs was observed at frequencies from 70 to 90 Hz than at frequencies from 30 to 50 Hz [68]. In some studies involving healthy participants, there was a general increase in muscle activity when the frequency of spinal cord stimulation increased from 5 to 40 Hz [20]. Similarly, more variability in the frequency of stimulation of the spinal cord was presented in the determination of the optimal stimulation frequency with the lowest intensity to produce the greatest muscle response [69].

Our studies revealed that tSCS with frequencies of 5 Hz and 30 Hz, both at subthreshold and suprathreshold intensities, reduced the center of pressure oscillations. Interestingly, at 5 Hz stimulation, the predominant signs of instability were observed in the open-eye trials (increased root-mean-square distance in the frontal plane and increased ellipse area), whereas at 30 Hz stimulation, they were observed in the closed-eye trials. These differences can be shown by analyzing the frequency modulation of the stabilogram spectrum, especially in tests on a soft surface. The tSCS at 5 Hz decreased the very low frequencies of the spectrum and increased the low range from 0.1 to 1 Hz, shifting 60% of the signal power into the vestibular range. The observed redistribution of the power spectrum indicates that our participants relied more on the vestibular/somatosensory and proprioceptive systems [59]. At 30 Hz tSCS, there was a distinct increase in the fluctuations of the high-frequency spectrum (over 2 Hz), probably supported by the continuous activity of the short-circuit reflexes [70]. It has been suggested that a higher frequency of cervical tSCS may provide improved voluntary control during stance [71]. Another mechanism may be the activation of cortical networks by cervical tSCS, facilitating corticospinal descending control [19]. Postural stability under these conditions may have been maintained by the rapid short movements provided by muscle contractions. Perhaps the faster muscle response during tSCS provided correction of postural fluctuations and resulted in better postural control during relaxed standing, which was reported in healthy participants [72]. The increase in postural stability shows that with cervical tSCS, the participants adopted a more successful balance control strategy. It is possible that the changes in control in the standing position were not directly caused by tSCS but may have been facilitated and modulated by sensory information projected on supraspinal and spinal circuits.

### 4.4. Limitations

One limitation of this study is the small number of participants, as we observed individualized responses to stimulation in our participants, which has been mentioned previously [38]. We also state that this study lacked a control group, with the same general protocol but without stimulation. Additional EMG studies on ankle and hip muscle activity would be very valuable, given their importance in postural control. Our observations help to elucidate the mechanisms contributing to the use of tSCS for further research in evaluating the efficacy and selection of optimal parameters of electrical stimulation to improve postural control in humans to improve motor function after neurological injury. It is possible that such methods could help maintain and improve postural stability in elderly individuals or optimize training in athletes. We have gathered extensive experimental material that cannot be adequately presented in a single article owing to its complexity and volume. This experience has been our first step in discussing the results obtained. However, we are confident that the results have opened significant opportunities for further analysis and interpretation, which requires more in-depth and detailed research.

## 5. Conclusions

This study revealed the effect of cervical tSCS on postural stability. The results demonstrate a positive effect of the tSCS on stability. This study also highlights the potential of using tSCS to improve stability in patients with movement disorders. However, it is necessary to conduct further research with control groups and rigorous protocols to assess the effectiveness of tSCS more accurately. Our study emphasizes the need for further research for developing effective rehabilitation techniques and improving postural stability in patients with movement disorders.

## Figures and Tables

**Figure 1 jfmk-09-00142-f001:**
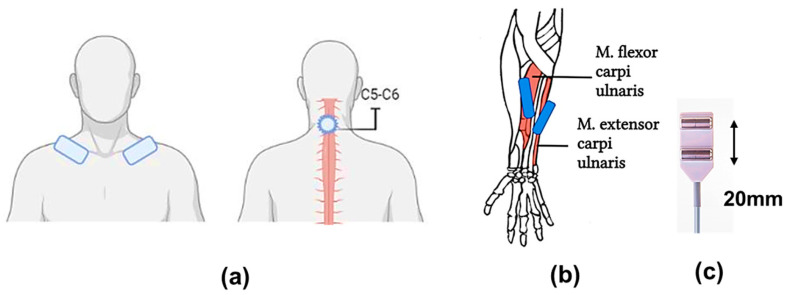
Experimental conditions: (**a**) tSCS: placement of the stimulating tip electrode (cathode) between the spinous processes of the C5 and C6 vertebrae and the rectangular electrode (anodes) symmetrically on the clavicles; (**b**) placement of surface electrodes for withdrawal of motor responses on the abdomen of the studied muscles: m. flexor carpi ulnaris, m. extensor carpi radialis; (**c**) surface electrodes with a fixed interelectrode distance of 20 mm.

**Figure 2 jfmk-09-00142-f002:**
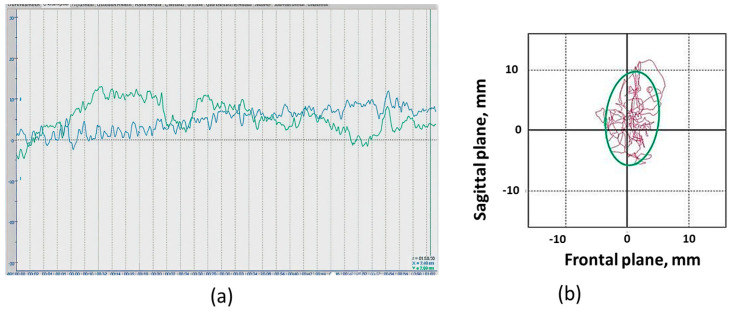
Recording examples: (**a**) Stabilogram of fluctuations in the center of pressure in the frontal plane (blue line) and in the sagittal plane (green line) of the “eyes open” position; (**b**) Example of a participant’s statokinesiogram. Statokinesiogram with mathematical processing of fluctuations in the center of pressure and construction of an ellipse describing 90% of the surface occupied by the statokinesiogram and reflecting the area of the subject’s support during the examination. The irregularly shaped line inside and outside the ellipse is the statokinesiogram. The side of the cell is equal to 10 mm.

**Figure 3 jfmk-09-00142-f003:**
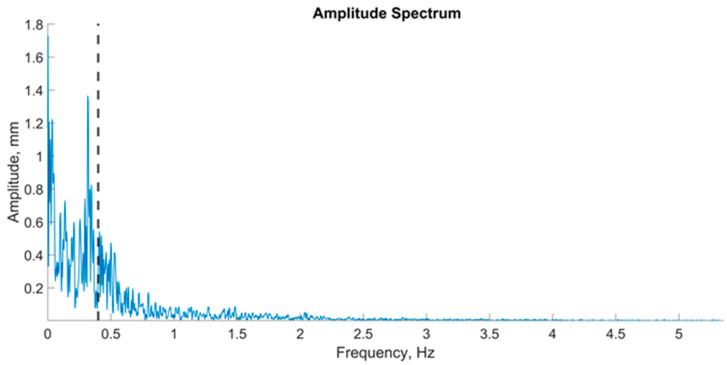
An example of a graph of the amplitude spectrum center of pressure oscillations in the frontal plane; the vertical dashed line on the graph is an indicator of 60% of the spectrum power.

**Figure 4 jfmk-09-00142-f004:**
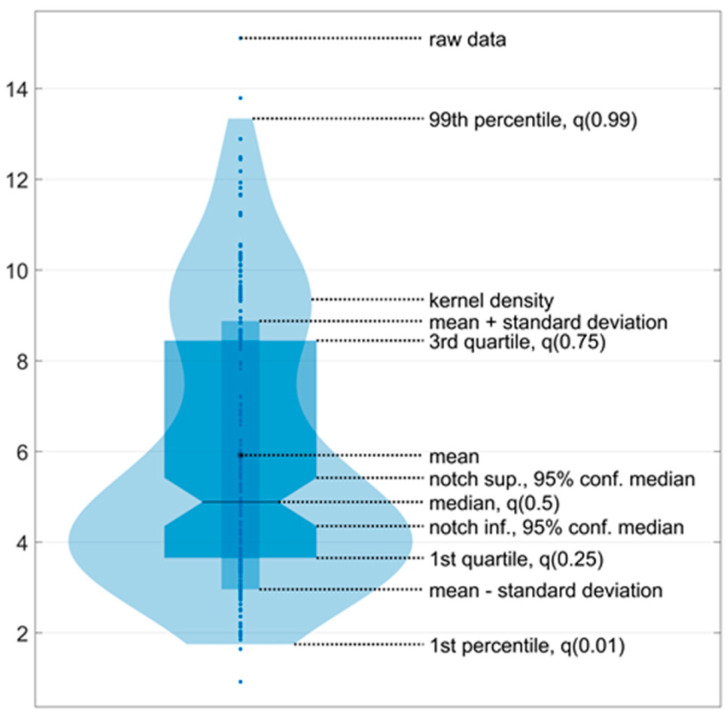
An example of a violin plot. The violin plots show the kernel probability density of the underlying data at different values. The violin plots include a marker for the median, the mean of the data, and a box indicating the interquartile range, as in standard box plots.

**Figure 5 jfmk-09-00142-f005:**
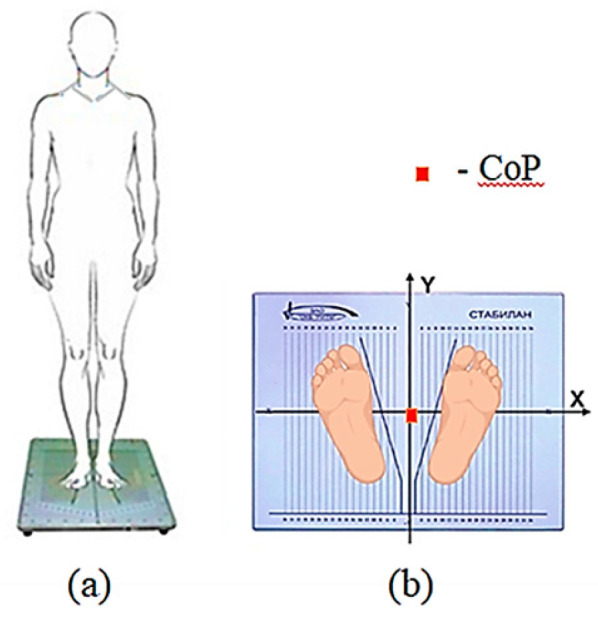
Experimental procedure: (**a**) Schematic of participants’ vertical position on the force plate; (**b**) Feet placement on the force plate (European position). The orientation of the force plate coordinate system relative to the participants’ sagittal plane is illustrated. Before testing, centering was performed—the position of the person’s center of pressure was aligned with the origin of the coordinates—indicated by the red square in the figure.

**Figure 6 jfmk-09-00142-f006:**
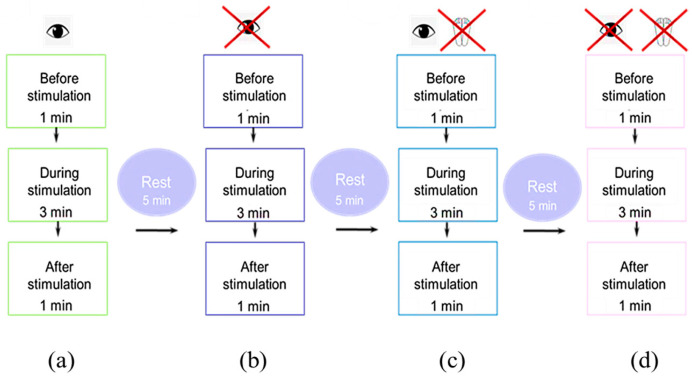
Schematic of the experiment in the (**a**) eyes-open trial; (**b**) eyes-closed trial; (**c**) soft surface trial with eyes open and (**d**) closed.

**Figure 7 jfmk-09-00142-f007:**
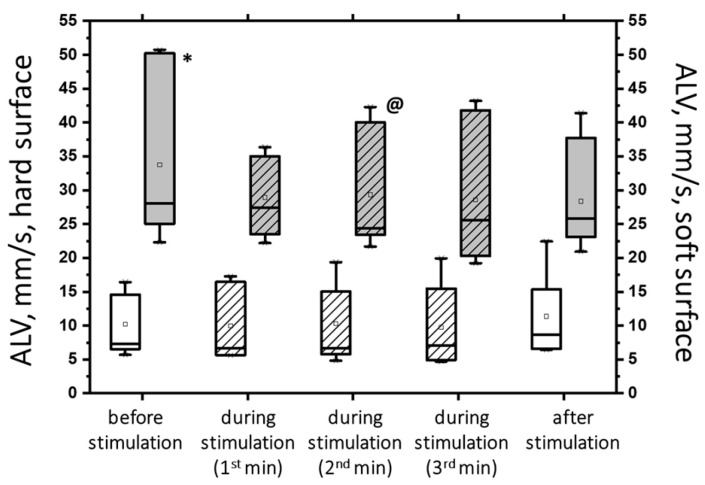
The average linear velocity (AVL) of the CoP displacement in the closed-eye test on a hard surface (white bars) and a soft surface (gray bars) before, during, and after transcutaneous cervical spinal cord stimulation at 5 Hz with subthreshold stimulus intensity. The data collected during stimulation were divided into three equal one-minute intervals (bars with slashes). The data are presented in box-and-whisker plots, where the boundaries of the boxes indicate the 1st and 3rd quartiles, and the whiskers extend 1.5 interquartile ranges above and below. The solid black lines indicate the median, and the dots indicate the mean; * *p* < 0.05 relative to the values on the hard surface; @ *p* < 0.05 relative to the values before stimulation.

**Figure 8 jfmk-09-00142-f008:**
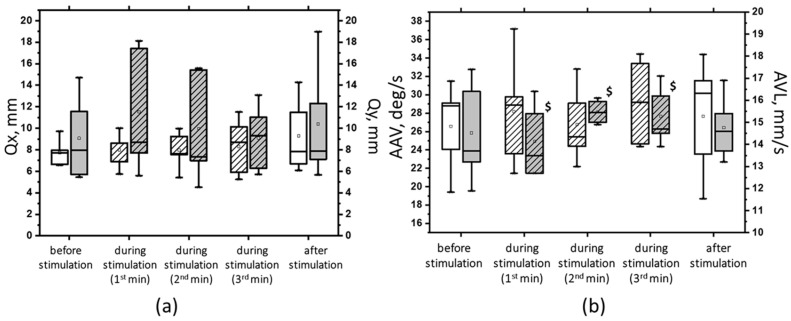
The root-mean-square deviation of the CoP along the frontal (Qx, white bars) and sagittal (Qy, gray bars) planes in the sample with eyes closed on a soft surface (**a**); the average linear velocity (ALV, white bars) and the average angular velocity (AAV, gray bars) of the deviation of the CoP in the sample with closed eyes on a soft surface (**b**) at 5 Hz suprathreshold tSCS; the data during stimulation were divided into three equal one-minute intervals (bars with slashes). The data are presented in box-and-whisker plots, where the boundaries of the boxes indicate the 1st and 3rd quartiles, and the whiskers extend 1.5 interquartile ranges above and below. The solid black lines indicate the median, and the dots indicate the mean; $—*p* < 0.05 vs. values of indices at subthreshold tSCS.

**Figure 9 jfmk-09-00142-f009:**
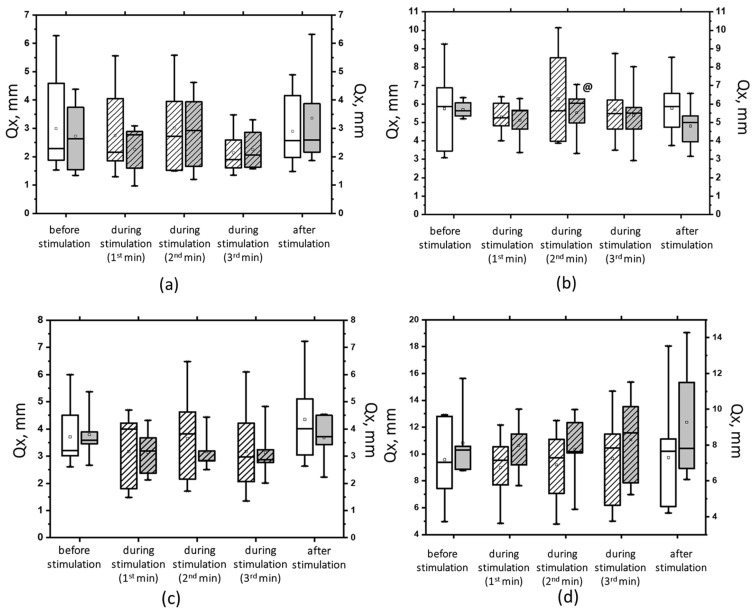
The root-mean-square deviation of the center of pressure along the frontal axis (Qx) in the eyes-open test on a hard (**a**) and a soft (**b**) surface and in the eyes-closed test on a hard (**c**) and a soft (**d**) surface before, during, and after tSCS at 5 Hz with subthreshold stimulus intensity (white bars) or suprathreshold stimulus intensities (gray bars). The data collected during stimulation were divided into three equal one-minute intervals (bars with slashes). The data are presented in box-and-whisker plots, where the boundaries of the boxes indicate the 1st and 3rd quartiles, and the whiskers extend 1.5 interquartile ranges above and below. The solid black lines indicate the median, and the dots represent the means. @ *p* < 0.05 relative to the values before stimulation.

**Figure 10 jfmk-09-00142-f010:**
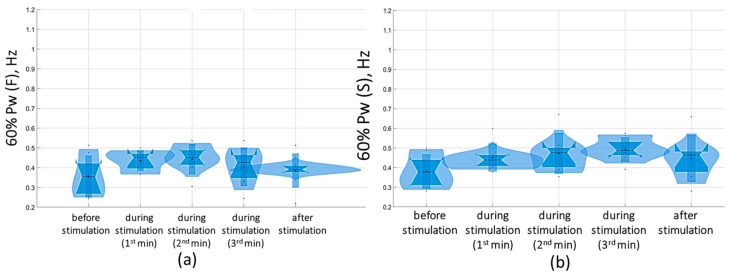
Sixty percent power of the stabilogram spectrum (60% Pw) in the sagittal (**a**) and frontal (**b**) planes before, during, and after tSCS at 5 Hz stimulation with the subthreshold intensity in a test with closed eyes on a hard surface; the data collected during stimulation were divided into three equal one-minute intervals; The violin plots show the kernel probability density of the underlying data at different values. The violin plots include a marker for the median, the mean of the data, and a box indicating the interquartile range, as in standard box plots.

**Figure 11 jfmk-09-00142-f011:**
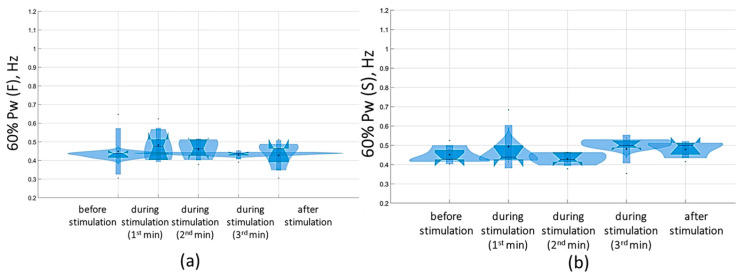
Sixty percent power of the stabilogram spectrum (60% Pw) in the sagittal (**a**) and frontal (**b**) planes before, during, and after tSCS at 5 Hz with the suprathreshold intensity in a test with closed eyes on a hard surface; the data collected during stimulation were divided into three equal one-minute intervals; the violin plots show the kernel probability density of the underlying data at different values. The violin plots include a marker for the median, the mean of the data, and a box indicating the interquartile range, as in standard box plots.

**Figure 12 jfmk-09-00142-f012:**
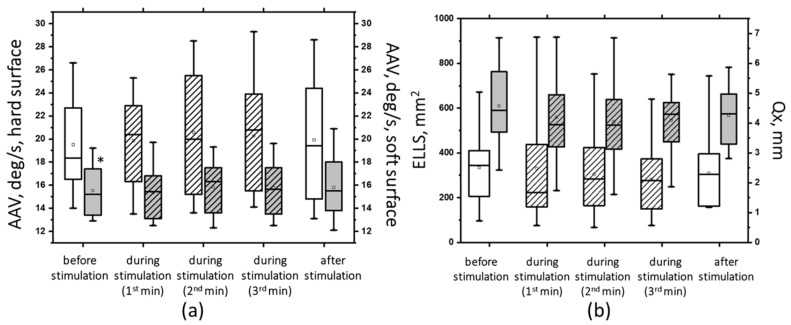
The average angular velocity (AAV) in the test with closed eyes on a hard (white bars) and soft (gray bars) surface (**a**); the area of the ellipse (ELLS, white bars) and the root-mean-square deviation of the CoP along the frontal plane (Qx, gray bars) in the test with closed eyes on a soft surface (**b**) before, during, and after tSCS at 30 Hz with subthreshold stimulus intensity; the data during stimulation were divided into three equal one-minute intervals (bars with lines). The data are presented in box-and-whisker plots, where the boundaries of the boxes indicate the 1st and 3rd quartiles, and the whiskers extend 1.5 interquartile ranges above and below. The solid black lines indicate the median, and the dots indicate the means; *—*p* < 0.05 relative to the values on the hard surface.

**Figure 13 jfmk-09-00142-f013:**
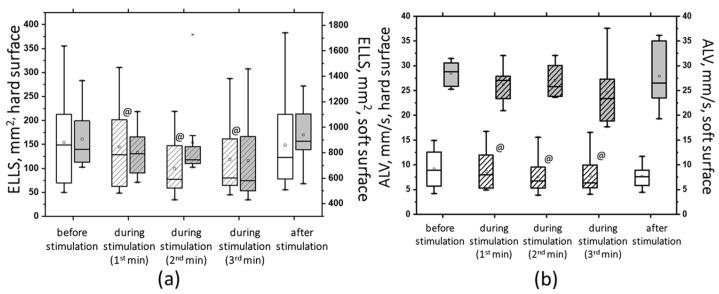
The area of ellipse (ELLS) (**a**) and the average linear velocity (ALV) (**b**) in the closed-eye test on a hard surface (white bars) and a soft surface (gray bars) before, during, and after tSCS at 30 Hz with subthreshold stimulus intensity; the data collected during stimulation were divided into three equal one-minute intervals (bars with slashes). The data are presented in box-and-whisker plots, where the boundaries of the boxes indicate the 1st and 3rd quartiles, and the whiskers extend 1.5 interquartile ranges above and below. The solid black lines indicate the median, and the dots indicate the mean; @—*p* < 0.05 relative to the values before stimulation.

**Figure 14 jfmk-09-00142-f014:**
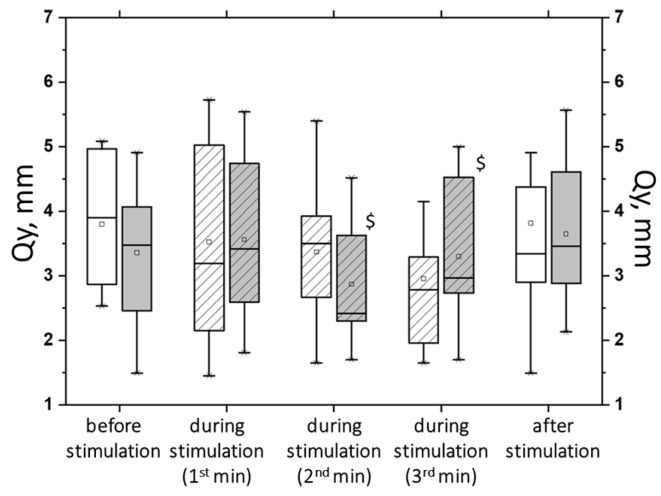
The root-mean-square deviation of the center of pressure along the sagittal plane (Qy) in the closed-eyes test on a hard surface before, during, and after tSCS at 30 Hz with subthreshold stimulus intensity (white bars) or suprathreshold stimulus intensities (gray bars); the data collected during stimulation were divided into three equal one-minute intervals (bars with slashes). The data are presented in box-and-whisker plots, where the boundaries of the boxes indicate the 1st and 3rd quartiles, and the whiskers extend 1.5 interquartile ranges above and below. The solid black lines indicate the median, and the dots represent the means. $—*p* < 0.05 relative to the values between the subthreshold and suprathreshold tSCS.

**Figure 15 jfmk-09-00142-f015:**
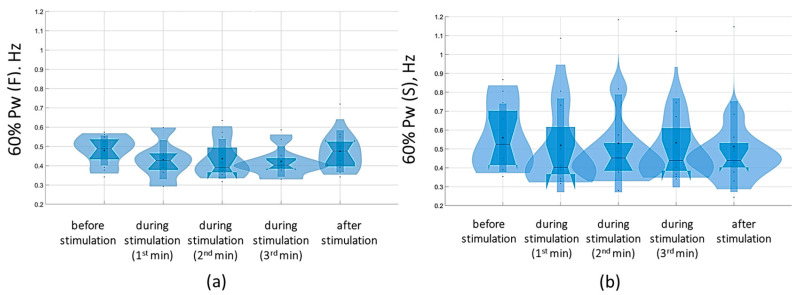
Sixty percent power of the stabilogram spectrum in the sagittal (**a**) and frontal (**b**) planes before, during, and after tSCS at 30 Hz with the subthreshold intensity in a test with closed eyes on a hard surface; the data collected during stimulation were divided into three equal one-minute intervals; the violin plots show the kernel probability density of the underlying data at different values. The violin plots include a marker for the median, the mean of the data, and a box indicating the interquartile range, as in standard box plots.

**Figure 16 jfmk-09-00142-f016:**
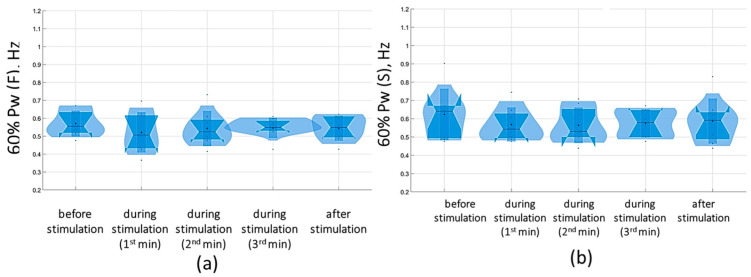
Sixty percent power of the stabilogram spectrum in the sagittal (**a**) and frontal (**b**) planes before, during, and after tSCS at 30 Hz with the suprathreshold intensity in a test with closed eyes on a soft surface; the data collected during stimulation were divided into three equal one-minute interval violin plots showing the kernel probability density of the underlying data at different values. The violin plots include a marker for the median, the mean of the data, and a box indicating the interquartile range, as in standard box plots.

## Data Availability

The initial data confirming the conclusions of this article will be provided by the authors upon request.

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
