# Peer review of "Different Factors Influencing Postural Stability during Transcutaneous Electrical Stimulation of the Cervical Spinal Cord"

_jfmk, 2024, doi:10.3390/jfmk9030142_

Round 1
Reviewer 1 Report
Comments and Suggestions for Authors
The primary purpose of this research was to examine the effect of subthreshold and suprathreshold transcutaneous electrical stimulation of the cervical spine of healthy individuals on standing postural stability under different conditions. General comments include a thorough English grammar edit for corrections in word tense, word choice, and paragraph coherence; the need for more detailed theoretical development of this method in the introduction; more clarity and technical information in the methods; significantly improved information, clarity, and coherence in the results and discussion sections. It appears that some of the references cited may not support the authors' claims appropriately. Specific comments can be found in the pdf.

corrections for errors in English grammar, eg, word choice and word tense, as well as paragraph coherence
Author Response
- Summary
Dear reviewer, thank you very much for taking the time to review this manuscript. We appreciate the helpful help you have provided, and your input will definitely help improve our manuscript. We apologize to the reviewer for the insufficient quality of the text and thank you for your comments to improve the quality of the article. Please find the detailed responses below and the corresponding revisions.
Comments 1: The primary purpose of this research was to examine the effect of subthreshold and suprathreshold transcutaneous electrical stimulation of the cervical spine of healthy individuals on standing postural stability under different conditions. General comments include a thorough English grammar edit for corrections in word tense, word choice, and paragraph coherence; the need for more detailed theoretical development of this method in the introduction; more clarity and technical information in the methods; significantly improved information, clarity, and coherence in the results and discussion sections. It appears that some of the references cited may not support the authors' claims appropriately. Specific comments can be found in the pdf.
Response 1: Yes, we have revised the manuscript, and corrected all the grammatical errors mentioned.
Comments 2: English grammar edit. I will highlight other word tense errors but please edit/correct entire manuscript for this issue.
Response 2: We have made all the corrections. Thank you:
Line 8 Science Department; Federal
Line 23; 393 stimulus strength
Line 24 predominated
Line 24; 25 increases
Lines 25; 26; 32; 63; 68; 158 is. Modifications have been implemented.
Comments 3: Line 32 is complicated process, should be "is a complex process..."; there may be many such grammatical errors which I will also highlight suggesting that expert English grammar edits are warranted.
Response 3: Line 33. You are right; it should be “is a complex process...” This has been changed. Thank you for pointing that out.
Comments 4: Line 39-43. More detailed background is needed in this section since it appears to provide a theoretical framework for the method used in this research
Response 4: Thank you; the following information has been added.
Line 41-49. The cervical spinal cord has been shown to contribute to fine control of movement because it contains many proprioceptors [Boyd-Clark, L. C.; Briggs, C. A.; Galea, M. P. Muscle spindle distribution, morphology, and density in the longus colli and multifidus muscles of the cervical spine. Spine. 2002, 27(7), 694–701. https://doi.org/10.1097/00007632-200204010-00005]. Cervical proprioception is particularly important for spatial orientation because it provides a frame of reference for how the head is positioned and moves relative to the trunk. While vestibular afferents encode active and passive head movements, cervical proprioception acts as a reference point for perceiving the actual position of the head relative to the trunk. This function is essential for the correct interpretation of visual and vestibular inputs, such as gaze stabilization during head movements [Campbell, D., Murphy, B. A., Burkitt, J., La Delfa, N., Sanmugananthan, P., Ambalavanar, U., & Yielder, P. (2023). Cervico-Ocular and Vestibulo-Ocular Reflexes in Subclinical Neck Pain and Healthy Individuals: A Cross-Sectional Study. Brain sciences, 13(11), 1603. https://doi.org/10.3390/brainsci13111603]. Proprioception is also important for the execution of coordinated, corrective and targeted intersegmental movements [Malmström, E.M.; Fransson, P.A.; Jaxmar Bruinen, T.; Facic, S.; Tjernström, F. Disturbed cervical proprioception affects the perception of spatial orientation while in motion. Experimental Brain Research 2017, 235, 2755-2766. https://doi.org/10.1007/s00221-017-4993-5]. The neuromuscular and sensory pathways of the cervical spinal cord and brain form connections that are important components for maintaining postural tone and neuromuscular coordination [6, 7]. The presence of such connections makes it possible to use tSCS to modulate descending pathways. In neurologically healthy participants, tonic activation of spinal cord networks through multiple sites of tSCS has been shown to facilitate both the spinal reflex and corticospinal pathways [Parhizi, B.; Barss, T. S.; Mushahwar, V. K. Simultaneous Cervical and Lumbar Spinal Cord Stimulation Induces Facilitation of Both Spinal and Corticospinal Circuitry in Humans. Frontiers in neuroscience. 2021, 15, 615103. https://doi.org/10.3389/fnins.2021.615103].
Comments 5: Line 47-52 In addition to showing that tSCS is useful in treating certain pathologies please present a brief theoretical framework for the use of this method (of course, with appropriate references)
Response 5: Thank you, we added the following information:
Line 58-66. tSCS, like epidural stimulation, is capable of altering excitability through the recruitment of posterior-root fibers to activate interneuronal networks, potentiate the generation of postsynaptic excitatory potentials and contribute to the excitation threshold of interneurons and motor neurons. This allows the spinal neuronal network to respond to the descending impulse and increases the overall excitability of the spinal network and potentially the motor cortex [Hofstoetter, U. S., Freundl, B., Binder, H., & Minassian, K. (2018). Common neural structures activated by epidural and transcutaneous lumbar spinal cord stimulation: Elicitation of posterior root–muscle reflexes. PloS one, 13(1), e0192013. https://doi.org/10.1371/journal.pone.0192013; Benavides, F. D., Jo, H. J., Lundell, H., Edgerton, V. R., Gerasimenko, Y., & Perez, M. A. (2020). Cortical and Subcortical Effects of Transcutaneous Spinal Cord Stimulation in Humans with Tetraplegia The Journal of neuroscience : the official journal of the Society for Neuroscience, 40(13), 2633-2643. https://doi.org/10.1523/JNEUROSCI.2374-19.2020 ]. However, unlike epidural stimulation, CESSM does not require surgical implantation of electrodes. A special configuration of pulses with carrier frequencies from 5 to 10 kHz minimises discomfort when using the CESSM [Singh G, Lucas K, Keller A, Martin R, Behrman A, Vissarionov S, Gerasimenko YP. Transcutaneous Spinal Stimulation From Adults to Children: A Review. Top Spinal Cord Inj Rehabil. 2023 Winter;29(1):16-32. doi: 10.46292/sci21-00084. ; Gorodnichev, R. M., Pivovarova, E. A., Pukhov, A., Moiseev, S. A., Savokhin, A. A., Moshonkina, T. R., Shcherbakova, N. A., Kilimnik, V. A., Selionov, V. A., Kozlovskaia, I. B., Edgerton, V. R., & Gerasimenko, I. uP. (2012). Fiziologiia cheloveka, 38(2), 46-56; Gerasimenko, Y., Gorodnichev, R., Moshonkina, T., Sayenko, D., Gad, P., & Reggie Edgerton, V. (2015). Transcutaneous electrical spinal-cord stimulation in humans. Annals of physical and rehabilitation medicine, 58(4), 225-231. https://doi.org/10.1016/j.rehab.2015.05.003].
Comments 6: Lines 72; 194 subjects Participants; please be consistent when referring to study volunteers
Response 6: Thank you for pointing this out. Therefore, we have replaced “subjects” with “participants”.
Line 92: 2.1. Participants of the study.
Comments 7: Lines 73 where from and how were participants recruited? and what were the specific inclusion criteria (you only provided very broad exclusion criteria)
Response 7: All our participants were volunteers. For clarification, we have provided more details about them and the selection criteria:
Line 93-99. Fourteen volunteers (3 males, 11 females, aged 18-27 years) were included in the study. The height of the participants was 163.71 ± 10.75 cm, and their weight was 59.29 ± 16.96 kg. All the participants rated themselves as healthy on the day of the study. The specific inclusion criteria were the absence of implanted metallic or electrical devices, pregnancy, and musculoskeletal or neurological disorders that could affect postural balance and well-being on the day of the study.
The anthropological data of all the participants are also in table form in the supplementary material.
Comments 8: Lines 76 World Medical Association
Response 8: We agree with this comment. Therefore, we have written the World Medical Association correctly.
Line101. The study was conducted with the informed voluntary consent of the participants in accordance with the Declaration of Helsinki developed by the World Medical Association. The study protocol was approved by the Local Ethical Committee of Kazan Federal University (protocol No. 34 of 27.01.2022).
Comments 9: Lines 81-82; 127; 134 flexor and extensor do not need to be capitalized
Response 9: Thank you for pointing this out. Currently, the flexor and extensor are not capitalised. Lines 115; 120-121; 204
Comments 10: Lines 82 suggest the word "elicited" since you don' want to use the word "determined" twice in the same sentence; another example of English grammar error
Response 10: We respectfully agree with this comment. We hope that the revised sentence meets with your approval.
Lines 203-206: The motor response threshold of the right and left forearm muscles, specifically the m. flexor carpi ulnaris and m. extensor carpi radialis, was elicited among the participants to determine the optimal stimulation intensity. Further details regarding this procedure can be found in paragraph 2.2.”
Comments 11: Lines 86 aligning the position of the person's center of pressure (CoP). The CoP is a calculated position and one cannot see it when standing on a force plate; typically, with force plate software we can see the vertical ground reaction force but don't conflate this with the CoP
Response 11: We believe that you are correct in your assumption that the center of pressure is a calculated position, which explains why it is not visible when standing on the force platform. However, we assure you that everything is written correctly, as our platform has a feature that allows us to do this. For clarity, we have swapped the points in the Methods section and renamed point 2.2 to point 2.5. In the new point 2.3 on stabilometry, we have added additional information about our equipment, which we hope will be useful. We present the description of the force platform as follows:
Lines 139-143: For stabilometry, the Stabilan-01-2 force platform system (Ritm LLC, Taganrog, Russia) with StabMed 2.13 software was used [Sliva S.S. Domestic computer stabilography: technical standards, functionality and areas of application. Biomed Eng 39, 31–34 (2005). https://doi.org/10.1007/s10527-005-0037-8]. The system recorded the position of the center of pressure (CoP) with a sampling frequency of 50 Hz and a resolution of <0.01 mm. For clarity, Figure 4 shows an example of a stabilogram and statokinesiogram recording.
On the stabilographic platform, on which a person is in a standing position, force sensors are attached that measure the reaction of the person's foot support. The two-coordinate computer stabilograph ensures the recording and analysis of the trajectory of the center of pressure exerted by a person on the horizontal working surface of the force-coordinate platform».
Comments 12: Lines 94 A total of 2 sessions
should be "two"; numbers below 10 should be written out unless part of a phrase, eg, "2-way"
Response 12: We agree with this comment. Therefore, we have rewritten the sentence.
"A total of two sessions with subthreshold and suprathreshold stimulus strengths were conducted on the same day, with a 15-minute interval between each session."
Comments 13: Lines 102 Conditional line; I am not familiar with the concept of a conditional line - what is it?
Response 13: We concur with the reviewer's assessment that this figure lacks informative value. The figure illustrates the position of the CoP on the platform; however, it is a replication of Figure 5b, and thus has been excluded.
Comments 14: Lines 138-144 Please describe the placement of surface electrodes; what was done to verify the accuracy of the placement of surface electrodes; if any prior calibration of the EMG system was done prior to testing; what the test-retest reliability of the EMG system is (with references); what frequency was used to collect muscle activity; what filtering of the EMG was done (with references); there are international standards for the use and reporting of kinesiological EMG which should be used and referenced
Response 14:
Lines 126-137: The electromyographic (EMG) activity was recorded and subsequently processed using a Neuro-MVP-8 electroneuromyograph (Neurosoft, Ivanovo, Russia), which is an 8-channel amplifier with Neuro-MVP software. The EMG data were filtered using a 50 Hz notch filter and a 20–1000 Hz bandpass filter. The data were recorded at a frequency of 4 kHz, exported, and analyzed with Neuro-MVP software. Surface electrodes (BE-1, NS990998.028, Neurosoft, Russia) were placed on the bellies of the right and left muscles: m. flexor carpi ulnaris and m. extensor carpi radialis; the interelectrode distance was 20 mm (Figure 3b, c), and the placement site was preliminarily determined by palpation. Muscle responses were recorded without changing the position of the EMG electrodes [Stålberg, E., van Dijk, H., Falck, B., Kimura, J., Neuwirth, C., Pitt, M., Podnar, S., Rubin, D. I., Rutkove, S., Sanders, D. B., Sonoo, M., Tankisi, H., & Zwarts, M. (2019). Standards for quantitative assessment of EMG and neurography. Clinical neurophysiology: official journal of the International Federation of Clinical Neurophysiology, 130(9), 1688–1729. https://doi.org/10.1016/j.clinph.2019.05.008; Besomi, M., Devecchi, V., Falla, D., McGill, K., Kiernan, M. C., Merletti, R., van Dieën, J. H., Tucker, K., Clancy, E. A., Søgaard, K., Hug, F., Carson, R. G., Perreault, E., Gandevia, S., Besier, T., Rothwell, J. C., Enoka, R. M., Holobar, A., Disselhorst-Klug, C., Wrigley, T., … Hodges, P. W. (2024). Consensus for experimental design in electromyography (CEDE) project: Checklist for reporting and critically appraising studies using EMG (CEDE-Check). Journal of electromyography and kinesiology: official journal of the International Society of Electrophysiological Kinesiology, 76, 102874. https://doi.org/10.1016/j.jelekin.2024.102874]. The threshold stimulation intensity for the occurrence of motor responses was assessed twice. The first test was conducted with 5 Hz stimulation, and the second test was conducted with 30 Hz stimulation on a different day.
Comments 15: Lines 145-148 Please describe any calibration or zeroing procedures of the force plate; where these procedures used just once or before each condition? What frequency was used to collect force data? Were raw force data processed, i.e., filtered, - if so, provide specific details in addition to providing the references for this selection. Since more than one session was conducted, please provide references attesting to the precision, i.e., test-retest reliability of the force plate
Response 15: Thank you for pointing this out. We have added information about ‘centering’ to section ‘2.3. Stabilography’
The possibility of “centering”, i.e., alignment of the coordinate system of the stability platform with the mathematical expectation of the subject's CoP, makes it possible to simplify the methods that do not require precise positioning of the subject on the stability platform. The system records the CoP positions with a sampling frequency of 50 Hz and a resolution of <0.01 mm. Before testing, centering was performed—the position of the person’s the center of pressure was aligned with the origin of coordinates.
Comments 16: Lines151-153 Figure 4. You failed to describe what "b" is
Response 16: Thank you for pointing this out. Therefore, we describe what Figure 2. "b" is
(b) Example of a participant's statokinesiogram. Line 146
Comments 17: Lines 159-174 This whole section is written/formatted incorrectly.
Response 17: Response 17: Thank you for pointing this out. Therefore, we have rewritten the paragraph.
Lines 159-165: When processing the statokinesiogram, the following static parameters were ana-lyzed: the area of the ellipse – an indicator characterizing 90% of the surface occupied by the statokinesiogram, reflecting the area of the subject's support during the examination (ELLS, mm²); the root mean square deviation of the CoP along the frontal (X) and sagittal (Y) axes (Qx and Qy, mm); and dynamic parameters: the average linear velocity of the CoP movement (ALV, mm/s); and the average angular velocity of the CoP movement (AAV, deg/s) (the formulas for the calculation are provided in Table S2).
Table S2. Analyzed center of pressure (CoP) parameters.
|
Parameter |
Definition |
Formula |
|
Confidence ellipse area, ELLS, mm2 |
The main part of the area occupied by the CoP without socalled loops and accidental outliers |
β – probability that the point of the statokinesiogram hits into the ellipse (β = 0.9). D(X), D(Y) – corresponding component dispersion |
|
RMSD along the frontal axis, Qx, mm |
Root mean square deviation of the CoP position along the frontal axis |
Xi– CoP coordinates in time N – number of counts |
|
RMSD along the sagittal axis, Qy, mm |
Root mean square deviation of the CoP position along the sagittal axis |
Yi – CoP coordinates in time N – number of counts |
|
ALV, mm/s |
The average linear velocity of the CoP movement, represented by the ratio of the length of the path of movement of the CoP to the duration of the test |
– instantaneous value of the velocity vector – experimental time |
|
AAV, deg/s |
The average angular velocity of the CoP movement – the average amplitude of the velocity of the participant's CoP during the duration of the examination |
– current change in the velocity vector angle – sampling time – number of velocity vectors |
Comments 18: Lines 176 should this be "center of pressure"? это должен быть «центр давления»?
Response 18: Yes, it is supposed to be "center of pressure".
Comments 19: Lines 183-184 Amplitude spectra of the fast Fourier transform with a Hamming window were found 183 for signals from the force platform. what software was used to determine this? was custom code written in Matlab or something similar? Please clarify this.
Response 19: Amplitude spectra of the fast Fourier transform with a Hamming window were found for signals from the force platform via a script developed in MATLAB R2019a.Line 249.
Comments 20: were data tested for normality, etc? please explain why nonparametric stats were used
Response 20: According to the authors, the hypothesis of normal distribution should not be applied. There were fewer than 15 participants. Therefore, nonparametric tests were performed. The kernel density plots we obtained (see all violin plots) illustrate this.
Comments 21: Line 212 static and dynamic parameters
I don't recall you providing lists of static and dynamic parameters or even operationally defining and differentiating what a "static" and "dynamic" parameter is; please clarify this (preferably in the methods section)
We are grateful for your observation.
The assessment of postural stability can be divided into two primary categories: (a) global posturographic parameters that assess the overall magnitude of the center of pressure (CoP) deviation and (b) structural posturographic parameters that describe the temporal nature of body sway. The former is traditionally used as an index of postural stability, considering CoP displacement as a manifestation of random sway. They are defined as static or, alternatively, as classical. The relationship between the magnitude of postural sway and the level of stability is incomplete without considering the dynamic properties of postural sway. To bridge this gap, nonlinear methods are used that describe the time-dependent structure of time series. In this context, vector indicators, which we define as dynamic, are used to gain insight into the mechanisms of postural instability.
In the Methods section, the parameters were delineated and classified into two categories: static and dynamic. Additionally, the stabilogram spectrum parameters were elucidated with the following explanations:
Lines 159–165: When the statokinesiogram was processed, the following static parameters were analyzed: the area of the ellipse—an indicator characterizing 90% of the surface occupied by the statokinesiogram, reflecting the area of the subject's support during the examination (ELLS, mm²); the root mean square deviation of the CoP along the frontal (X) and sagittal (Y) axes (Qx and Qy, mm); and the dynamic parameters—the average linear velocity of the CoP movement (ALV, mm/s) and the average angular velocity of the CoP movement (AAV, deg/s) (the formulas for the calculation are provided in Table S2).
Comments 22: Line 215 Figure 7.
- there are two lines connecting the bars - what are these?
- are these data means and SD?
- what do the oblique lines in the bars represent?
the information I am asking for should be provided in the figure caption
Response 22: You are correct, and we have expanded the figure legends.
1) The line connecting the bars represents the median value. We have removed it to avoid cluttering the figure;
2) The data are presented in a box-and-whisker plot, where the boundaries of the box indicate the 1st and 3rd quartiles, and the whiskers extend 1.5 interquartile ranges above and below. The solid black line indicates the median, and the dot represents the mean.
3) Bars with lines indicate the parameters during transcutaneous stimulation of the cervical spinal cord. Additionally, the data collected during stimulation were divided into three equal one-minute intervals.
We have included the answers to your questions in the legends:
Lanes 266–273: Figure 7. The average linear velocity of the center of pressure displacement in the closed-eye test on a hard surface (white bars) and a soft surface (gray bars) before, during, and after transcutaneous cervical spinal cord stimulation at 5 Hz with subthreshold stimulus intensity. The data collected during stimulation were divided into three equal one-minute intervals (bars with lines). The data are presented in a box-and-whisker plot, where the boundaries of the box indicate the 1st and 3rd quartiles, and the whiskers extend 1.5 interquartile ranges above and below. The solid black line indicates the median, and the dots indicate the mean; * - p<0.05 relative to the values on the hard surface; @ - p<0.05 relative to the values before stimulation.
Comments 23: Line 221-223 As can be seen in Figure 7 average linear velocity increased under conditions of increasing sample complexity but decreased under subthreshold tSCS at 2 min of stimulation. Also, the change in average linear velocity demonstrated an improvement in stability.
Response 23: We agree with the assertion made in your comment. The figures were scaled to facilitate visualization of the stimulation effects. This was seemingly an ill-advised decision. The scale has now been restored to its original state, thus allowing for the differences between experimental conditions to be readily discerned. Figure 7 and the accompanying text are presented here:
Lines 258–273: While some changes were statistically significant, this was not the case for all changes when 5 Hz transcutaneous electrical spinal cord stimulation (tSCS) was used at subthreshold and suprathreshold stimulus strengths (see Supplementary Material, Table S3). Clearly, 5 Hz tSCS improved postural stability in the eyes-closed soft surface (ECSS) condition. Most changes in the static and dynamic parameters of the center of pressure (CoP) oscillation demonstrated this improvement. For example, subthreshold tSCS stimulation decreased the average linear velocity (ALV) (Figure 7).
Figure 7 shows that ALV increased more than threefold (p<0.05) under the challenging test conditions (hard/soft surface). However, with subthreshold tSCS at 2 minutes of stimulation, the ALV decreased by 13% (p<0.05). Additionally, the average angular velocity (AAV) decreased by 33% under ECSS conditions (p < 0.05) and increased by 4% under stimulation (p = 0.36), although this difference was not statistically significant.
At 5 Hz, the tSCS of the suprathreshold intensity resulted in a clear shift toward stability under the conditions of ECSS. This was evidenced by an increase in ELLS, Qx, and Qy, which was not statistically significant (Figure 8a). Additionally, the dynamic parameters demonstrated a notable decrease in the ALV by 12% (p<0.05) and an increase in the AAV by 10% (p>0.05). This was a significant change compared with the values at subthreshold stimulus strength (Figure 8b).
However, it should be noted that at 5 Hz tSCS, regardless of the stimulation intensity, an increase in the root mean square distance in the frontal plane with open eyes was observed (Figure 9).
Comments 24: Line 227 Figure 8 appears to only show data for the eyes open condition and does not compare data to eyes closed condition so how can the reader do their own comparison?
Response 24: We agree with the reviewer's comment. We did not present the data for the closed-eyes condition, as no reliable changes were found in this condition. All the parameters are presented in Table 3 of the supplementary materials. We have added the Qx data for the closed-eyes test to the figure. Furthermore, we have relocated this information to the end of the section, as we believe it is more appropriate. It is now referred to as Figure 9, and the figure caption reads as follows:
Lines 299 – 307. Figure 9. The root mean square deviation of the center of pressure along the frontal axis (Qx) in the eyes-open test on a hard (a) and a soft (b) surface and in the eyes-closed test on a hard (c) and a soft (d) surface before, during, and after transcutaneous cervical spinal cord stimulation at 5 Hz with subthreshold stimulus intensity (white bars) or suprathreshold stimulus intensities (gray bars). The data collected during stimulation were divided into three equal one-minute intervals (bars with lines). The data are presented in a box-and-whisker plot, where the boundaries of the box indicate the 1st and 3rd quartiles, and the whiskers extend 1.5 interquartile ranges above and below. The solid black line indicates the median, and the dots represent the means; * p<0.05 relative to the values on the hard surface; @ p<0.05 relative to the values before stimulation.
Comments 25: Line 236 as noted in previous figures/captions the reader needs more explanatory information in the text and caption related to figure 9
Response 25: Yes, thank you. It will be Figure 8 now.
Lines 288-296. Figure 8. The root mean square deviation of the CoP along the frontal (Qx, white bars) and sagittal (Qy, gray bars) planes in the sample with eyes closed on a soft surface (a); the average linear velocity (ALV, white bars) and the average angular velocity (AAV, gray bars) of the deviation of the CoP in the sample with closed eyes on a soft surface (b) at 5 Hz suprathreshold tSCS; the data during stimulation were divided into three equal one-minute intervals (bars with lines). The data are presented in a box-and-whisker plot, where the boundaries of the box indicate the 1st and 3rd quartiles, and the whiskers extend 1.5 interquartile ranges above and below. The solid black line indicates the median, and the dot indicates the mean; $ - p<0.05 vs. values of indices at subthreshold tSCS.
Comments 26: Line 242-243 one sentence paragraphs are not acceptable; furthermore, it is not clear that this statement is in reference to, i.e, is there a figure?
Response 26: This paragraph presents the details of the spectral analysis of the stabilogram. We have combined the paragraphs and added a link to the table:
Thanks for pointing this out. This paragraph presents the details of the spectral analysis of the stabilogram. We have combined the paragraphs and added a link to the table. The numbers have also been added. The text now reads:
Lines 308-316: Overall, with subthreshold 5-Hz tSCS, 60% of the spectral power (60% Pw) was 0.4 [0.3; 0.5] Hz in the sagittal plane and 0.3 [0.3; 0.5] Hz in the frontal plane, corresponding to the low-frequency range (Pw2) that characterizes the predominant influence of the vestibular system. With eyes closed, 5 Hz subthreshold tSCS resulted in a shift of 60% Pw to-ward the mid-frequencies (Pw3): 0.5 [0.4; 0.5] Hz and 0.4 [0.4; 0.5] Hz in the sagittal and frontal planes, respectively (Figure 10). This shift was due to an increase in the proportion of Pw3 to 32.7 [29.5; 36.2] % and Pw4 (high frequencies) to 9 [8.1; 9.3] % in the sagittal plane; in the frontal plane, the proportion of Pw3 increased to 31 [25.6; 34.7] %, Pw4 to 7.2 [6.2; 8.9] % (Table S5)".
Comments 27: Line 246 PW please write this out with PW placed in parentheses
Response 27: You were right to bring this to our attention. This change has been made.
Comments 28: Line 247 Pw3 and Pw4 although these might be shown in supplementary material they need clear definition here, i.e, write out what Pw3 and Pw4 mean
Response 28: You are correct in pointing this out. We have added spectrum parameters with decoding to the methods.
Lines 167–177: Amplitude spectra of the fast Fourier transform with a Hamming window were found for signals from the force platform via a script developed in MATLAB R2019a [34] (Figure 3). The frequency was plotted on the abscissa axis, and the region of interest was defined as 0.02–5 Hz. The signal amplitude was plotted on the vertical (ordinate) axis. The frequency corresponding to 60% of the power was identified at the point corresponding to 60% of the signal area across the entire frequency range of 0.02–5 Hz. This value represented 60% of the power of the stabilogram spectrum, which was designated 60%Pw. The power spectrum was divided into four frequency ranges: the ultralow frequency range from 0.02 to 0.2 Hz (Pw1), the low frequency range from 0.1 to 1.0 Hz (Pw2), the midfrequency range from 0.5 Hz to 2 Hz (Pw3), and the high frequency range from 2 Hz to 5 Hz (Pw4).
Comments 29: Line 250 Figire 10 more explanation is needed in the methods/stats section and the figure caption in terms of what the reader is looking at, i.e., what kind of analysis was done and what is the reader looking at in the figure
Response 29: Yes, we agree with you. In section 2.4 of the research methods, we have included a description of a violin plot and provided an example of a drawing:
Lines 192–194: Figure 4. Example of a violin plot. Violin plots show the kernel probability density of the underlying data at different values. Violin plots include a marker for the median, the mean of the data, and a box indicating the interquartile range, as in standard box plots.
Lines 318-323: Figure 10. Sixty percent power of the stabilogram spectrum (60% Pw) in the sagittal (a) and frontal (b) planes before, during, and after tSCS at 5 Hz stimulation with subthreshold intensity in a test with closed eyes on a hard surface; the data collected during stimulation were divided into three equal one-minute intervals; violin plots show the kernel probability density of the underlying data at different values. Violin plots include a marker for the median, the mean of the data, and a box indicating the interquartile range, as in standard box plots.
Comments 30: Line 260 same concerns regarding a one-sentence paragraph, lack of detail in the text and the figure 11 caption
Response 30: Yes, we agree with you; we have combined all the paragraphs and changed the caption to a figure:
Lines 324-328: At the suprathreshold tSCS intensity, the 60% Pw range shifted to 0.4 [0.4; 0.5] Hz in the sagittal plane and 0.5 [0.4; 0.5] Hz in the frontal plane in the eyes-closed test (Figure 11). This shift was attributable to an increase in the proportion of Pw4(S) to 7.9% [7.9; 8.2%] (Figure 11, supplementary material, Table S5). In general, during suprathreshold tSCS, the vestibular component prevailed in the spectrum, and the muscular component was added.
Lines 331-336: Figure 11. Sixty percent power of the stabilogram spectrum (60% Pw) in the sagittal (a) and frontal (b) planes before, during, and after tSCS at 5 Hz with suprathreshold intensity in a test with closed eyes on a hard surface; violin plots show the kernel probability density of the underlying data at different values. Violin plots include a marker for the median, the mean of the data, and a box indicating the interquartile range, as in standard box plots.
Comments 31: Line 273 Which is indicative of an enhanced ankle-brachial strategy.
I am not sure what this is. You did not present this idea in the introduction and you have not defined the "ankle-brachial strategy. Furthermore, explanations are usually saved for the discussion when you focus on important results, attempt to provide explanations and/or discuss related topics from the literature. So, I am not sure this statement is appropriate here.
Response 31: You are absolutely right. Thank you for pointing this out. We have removed this conclusion from the results.
Comments 32: Line 317 I have the same concerns regarding the text, figures and figure captions in all of sectio 3.2
Response 32: We hope that we have made the changes you suggested to the text, figures, and figure captions. We have followed the instructions in section 3.1 to the best of our ability. We have made all the corrections to the manuscript.
Comments 33: Line 329 I am not sure that the intent and information of ref#23 substantiates your broad statement here.
Response 33: We agree that our reference may be narrowly focused. We have added a reference to the research: Peterka R. J. (2018). Sensory integration for human balance control. Handbook of clinical neurology, 159, 27-42. https://doi.org/10.1016/B978-0-444-63916-5.00002-1, line 430.
Comments 34: Line 333-336 This section needs the integration of the one sentence paragraphs into a single coherent paragraph which appears to compare the overall results to previous research.
Response 34: Yes, you are correct; we have combined all the paragraphs into one.
Comments 35: Line 357 ref#5 is a scoping review reporting on the associatio between contact sport exposure and cervical sensorimotor dysfunction and NOT a report about normal cervical reflexes. Authors need consider a more appropriate reference here.
Response 35: We have added references to characterize the interaction of different sensory systems in postural control, although these too do not always purely describe normal reflexes as a function is defined when it changes: Armstrong, B., McNair, P., & Taylor, D. (2008). Head and neck position sense. Sports medicine (Auckland, N.Z.), 38(2), 101-117. https://doi.org/10.2165/00007256-200838020-00002; Artz NJ, Adams MA, Dolan P. Sensorimotor function of the cervical spine in healthy volunteers. Clin Biomech (Bristol, Avon). 2015 Mar;30(3):260-8. doi: 10.1016/j.clinbiomech.2015.01.005. Epub 2015 Jan 28. PMID: 25686675; PMCID: PMC4372261; Campbell, D., Murphy, B. A., Burkitt, J., La Delfa, N., Sanmugananthan, P., Ambalavanar, U., & Yielder, P. (2023). Cervico-Ocular and Vestibulo-Ocular Reflexes in Subclinical Neck Pain and Healthy Individuals: A Cross-Sectional Study. Brain Sciences, 13(11), 1603. https://doi.org/10.3390/brainsci13111603, line 455.
Comments 36: Line 363-367 The authors are not clear in this paragraph about the connection between these references and their primary results. More clarification is needed.
Response 36: You are correct that this information is now out of place. We have changed the paragraphs as follows:
Lines 461-470: As mentioned above, considering that postural control is not a simple summation of static reflexes but rather a complex skill based on the interaction of dynamic sensorimotor processes [39], we suggest that additional activation of the cervical spinal cord by tSCS, which changes the activity of both vestibular input and descending propriospinal tracts, may have a positive effect on postural control. The results revealed that the cervical spinal cord is activated by both rhythmic arm rotation [48, 49] and that tonic tSCS modulates the activity of lumbar networks [50]. This connection improves walking function after neurological impairment [12, 51]. These findings suggest that engaging cervical spinal cord networks through tSCS may not only improve cervical‒lumbar connectivity but also improve the ability to stand and walk [26].
Comments 37: Line 393 force
Response 37: Changed
Comments 38: Line 405 ref#43 did not examine the role of the thigh or trunk muscles associated with postural sway/tilt. Please recheck your references in this regard
Response 38: Yes, this article is not about muscle work; rather, it is about strategies for maintaining postural stability, in concept. The hip strategy is one of the leading strategies for decreased stability; in this case, the reference is reasonable in our opinion.
Comments 39: ALV
Response 39: "Average linear velocity".
Comments 40: Line 425 при
Response 40: Corrected
Comments 41: Line 443 this reference is a report on the reweighting of systems in patients with cervical myelopathy following surgery NOT about changes in normal healthy individuals. I am not sure this reference is appropriate
Response 41: Yes, this is a link to an article on pathological changes in postural control. Spectral analysis of body oscillations can also be useful in identifying various sources of imbalance in people with balance problems. This reference may be valid because it shows that the high-frequency spectrum of the stabilogram reveals features of postural stability maintenance in neurological pathology. However, we have added a reference to another source: Singh, N. B., Taylor, W. R., Madigan, M. L., & Nussbaum, M. A. (2012). The spectral content of postural sway during quiet stance: influences of age, vision and somatosensory inputs. Journal of electromyography and kinesiology : official journal of the International Society of Electrophysiological Kinesiology, 22(1), 131-136. https://doi.org/10.1016/j.jelekin.2011.10.007. Line 515.
Comments 42: Line 447 faster muscle response
this a speculative phrase since you did not measure muscle activity of muscles typically associated with postural control, eg, ankle and hip muscles. So what are you referring to here? Please clarify.
Response 42: It is believed that high-frequency oscillations are associated with the activity of the shortest neuronal circuits, reflecting the participation of muscle tone and proprioceptive information from muscles in the regulation of posture [Paillard, T., Bizid, R., & Dupui, P. (2007). Do sensorial manipulations affect subjects differently depending on their postural abilities? British journal of sports medicine, 41(7), 435–438. https://doi.org/10.1136/bjsm.2006.032904]. Therefore, spectral analysis of the stabilographic signal allows us to provide some indirect information about the predominant sensory contribution to pose regulation and about the preferred mechanisms (neuronal rings) in pose regulation. Moreover, it is necessary to determine the relative value of such interpretations of spectral indices.

Reviewer 2 Report
Comments and Suggestions for Authors
Dear authors,
The study shows the postural effects of applying transcutaneous electrical stimulation in the cervical spinal cord to healthy subjects in different postural input conditions. The design of this study is not clear due to the application of an intervention to participants and the lack of a clinical trial register. Authors should explain the type of study in methods section to clarify the need of register.
Lines 39-43: authors must explain further the mechanisms that allow cervical spine to influence in vestibular and proprioceptive systems. The description is vague and poorly detailed.
Line 80: section 2.2.: point 1) makes no sense without previous explanation. Maybe something like “was placed” is missing.
Lines 88-93: it is not clear if all test are performed to all people or if there are different groups.
Methods section: information about how recruitment was done is missing.
Figure 1: a) was it the meaning of the blue line at the neck of the participant? Are the electrodes? State in the figure. B) how many degrees are between the feet? c) what is the meaning of “conditional line” and what is its relation with the center of pressure? Information is missing.
Line 107: was it a rest between sessions? Did authors take into account the possible influence of the previous intervention in the next one?
Line 119: what was the size of the electrodes?
Figure 3: why are electrodes placed in that way?
Line 131: when was 5 Hz and when 30 Hz? As they cannot be both at the same time.
Figure 6: explanations of the meaning of “i”, “S”, “trapz” are missing. Unless authors provide further explanations, it is not clear the utility of the figure.
Line 201: indicate version of the software and manufacturer.
Line 205: groups or interventions? It seems like there are no groups, the same participants for every intervention.
Figure 13: Check if authors intended to mean “surfase” or “surface”. Please correct.
Line 324: in what way authors considered that postural intability benefit from tSCS? Regarding participants are healthy people, what condition is promoting the tSCS?
Line 454: I suggest authors to consider follow-up of the results as a limitation, anyway it is just a suggestion.
Author Response
- Summary
Thank you very much for taking the time to review this manuscript. The authors would like to express their gratitude to the reviewer for their meticulous reading of the work and the constructive comments they provided. In accordance with the recommendations set forth by the esteemed reviewer, the authors have implemented the following changes to the article:
Comments 1: The study shows the postural effects of applying transcutaneous electrical stimulation in the cervical spinal cord to healthy subjects in different postural input conditions. The design of this study is not clear due to the application of an intervention to participants and the lack of a clinical trial register. Authors should explain the type of study in methods section to clarify the need of register.
Response 1: Lines 101-103. The study was conducted with the informed voluntary consent of the participants in accordance with the Declaration of Helsinki developed by the World Medical Association. The study protocol was approved by the Local Ethical Committee of Kazan Federal University (protocol No. 34 of 27.01.2022). All the participants provided written informed consent.
This scientific research is the dissertation of Leysan Bikchentayeva. Accordingly, in addition to the necessary review by the local ethics committee of the university, the work was discussed at a meeting of the Academic Council of the Institute of Fundamental Medicine and Biology and was approved.
Comments 2: Lines 39-43: authors must explain further the mechanisms that allow cervical spine to influence in vestibular and proprioceptive systems. The description is vague and poorly detailed.
Response 2: Thank you; the following information has been added.
Line 41-55. The cervical spinal cord has been shown to contribute to fine control of movement because it contains many proprioceptors [Boyd-Clark, L. C.; Briggs, C. A.; Galea, M. P. Muscle spindle distribution, morphology, and density in the longus colli and multifidus muscles of the cervical spine. Spine. 2002, 27(7), 694–701. https://doi.org/10.1097/00007632-200204010-00005]. Cervical proprioception is particularly important for spatial orientation because it provides a frame of reference for how the head is positioned and moves relative to the trunk. While vestibular afferents encode active and passive head movements, cervical proprioception acts as a reference point for perceiving the actual position of the head relative to the trunk. This function is essential for the correct interpretation of visual and vestibular inputs, such as gaze stabilization during head movements [Campbell, D., Murphy, B. A., Burkitt, J., La Delfa, N., Sanmugananthan, P., Ambalavanar, U., & Yielder, P. (2023). Cervico-Ocular and Vestibulo-Ocular Reflexes in Subclinical Neck Pain and Healthy Individuals: A Cross-Sectional Study. Brain sciences, 13(11), 1603. https://doi.org/10.3390/brainsci13111603]. Proprioception is also important for the execution of coordinated, corrective and targeted intersegmental movements [Malmström, E.M.; Fransson, P.A.; Jaxmar Bruinen, T.; Facic, S.; Tjernström, F. Disturbed cervical proprioception affects the perception of spatial orientation while in motion. Experimental Brain Research 2017, 235, 2755-2766. https://doi.org/10.1007/s00221-017-4993-5]. The neuromuscular and sensory pathways of the cervical spinal cord and brain form connections that are important components for maintaining postural tone and neuromuscular coordination [6, 7]. The presence of such connections makes it possible to use tSCS to modulate descending pathways. In neurologically healthy participants, tonic activation of spinal cord networks through multiple sites of tSCS has been shown to facilitate both the spinal reflex and corticospinal pathways [Parhizi, B.; Barss, T. S.; Mushahwar, V. K. Simultaneous Cervical and Lumbar Spinal Cord Stimulation Induces Facilitation of Both Spinal and Corticospinal Circuitry in Humans. Frontiers in neuroscience. 2021, 15, 615103. https://doi.org/10.3389/fnins.2021.615103].
Comments 3: Line 80: section 2.2.: point 1) makes no sense without previous explanation. Maybe something like “was placed” is missing.
Response 3: The organization of the experiment has been relocated to section 2.5, and a supplementary clarification has been incorporated at the conclusion of the sentence, with a cross-reference to the paragraph that provides a more comprehensive account.
Lines 203-206: The motor response threshold of the right and left forearm muscles, specifically the m. flexor carpi ulnaris and m. extensor carpi radialis, was elicited among the participants to determine the optimal stimulation intensity. Further details regarding this procedure can be found in paragraph 2.2.
Comments 4: Lines 88-93: it is not clear if all test are performed to all people or if there are different groups.
Response 4: We agree with the reviewer that the experimental design is not adequately delineated.
In response to this question, we confirmed that each participant completed all stages of the experiment. The study was conducted without specific groups, with 14 volunteers who were examined in various tests.
This part of the description is like this (Lines 219-232):
The stabilography test was conducted under the following conditions for all participants: (1) a 1-minute trial without stimulation; (2) for 3 minutes with stimulation; and (3) for 1 minute without stimulation immediately after stimulation. In each session, we conducted a total of four tests (Figure 6). These were as follows: standing with eyes open (Figure 6a); standing with eyes closed (Figure 6b); standing on a soft, unstable support (foam pad 49 cm L × 49 cm W × 18 cm H with foot position marks) with eyes open (Figure 6c); and standing with eyes closed (Figure 6d). Each session lasted a maximum of 90 minutes, including setup, calibration, and testing. We took five-minute breaks between recordings to minimize fatigue. The participants were allowed to walk or rest on the force platform between recordings.
Comments 5: Methods section: information about how recruitment was done is missing.
Response 5: We agree with the reviewer. All of our participants were volunteers. For clarification, we have included more information about them and the selection criteria.
Line 93-99. Fourteen volunteers (3 males, 11 females, aged 18-27 years) were included in the study. The height of the participants was 163.71 ± 10.75 cm, and their weight was 59.29 ± 16.96 kg. All the participants rated themselves as healthy on the day of the study. The specific inclusion criteria were the absence of implanted metallic or electrical devices, pregnancy, and musculoskeletal or neurological disorders that could affect postural balance and well-being on the day of the study. The anthropological data of all the participants are also in table S1 in the supplementary material.
Comments 6: Figure 1: a) was it the meaning of the blue line at the neck of the participant? Are the electrodes? State in the figure. B) how many degrees are between the feet? c) what is the meaning of “conditional line” and what is its relation with the center of pressure? Information is missing.
Response 6: a) You are right; the blue line on the participant's neck is misleading. We have removed it from the figure. This picture illustrates the procedure of the experiment.
- b) We added more information about the subjects' leg position:
Lines 207-212: The participants were asked to stand on the force platform in a neutral position, with their arms at their sides and their feet in a heel together, toes apart position (commonly referred to as the “European position,” with heels positioned 2 cm apart and toes at an angle of 30°), aligning the sagittal plane with the anteroposterior axis of the force plate (Figure 5a, b). Centering was performed prior to testing to ensure that the person’s center of pressure was aligned with the origin.
- c) This picture was given to show the location of the CoP on the platform, but it duplicates picture ‘b’, so we removed it. Therefore, the caption for Figure 5 is as follows (line):
Figure 5. Experimental procedure: (a) Schematic of participants’ vertical position on the force plate; (b) Feet placement on the force plate (European position).The orientation of the force plate coordinate system relative to the participants' sagittal plane is illustrated.
Comments 7: Line 107: was it a rest between sessions? Did authors take into account the possible influence of the previous intervention in the next one?
Response 7: There was a rest of 5 minutes between trials. Stimulation at 5 Hz or 30 Hz was performed on different days. We have added detailed descriptions.
Lines 229-232: Each session lasted a maximum of 90 minutes, including setup, calibration, and testing. We took five-minute breaks between recordings to minimize fatigue. Participants were allowed to walk or rest on the force platform between recordings.
Lines 233-242: We conducted the study under the following conditions: open eyes (EO)/closed eyes (EC) and hard surface (HS)/soft surface (SS) in different combinations: EOHS; ECHS; EOSS; and ECSS. The order of the four recordings was randomized. After a short break (≤5 min), another random order of these four recordings followed. A total of two sessions with subthreshold and suprathreshold stimulus strengths were performed on one day, with 15 minutes in between. Each set of these four conditions was considered an independent series as a test. Thus, we obtained two measurements for each participant in all experimental conditions.
Comments 8: Line 119: what was the size of the electrodes?
Response 8: In this study, a five-channel BIOSTIM-5 stimulator (Cosyma Ltd., Moscow, Russia) was employed. A stimulating cutaneous round electrode (cathode) 32 mm in diameter with an adhesive layer (PG479/32, Fiab, UK) was positioned on the skin between the spi-nous processes of the C5 and C6 vertebrae. Rectangular electrodes (anodes) 45 × 80 mm in size with an adhesive layer (PG472W, Fiab, UK) were placed symmetrically on the clavi-cles (Figure 1A) [29]. As presented in the manuscript, Lines 105-110.
Comments 9: Figure 3: why are electrodes placed in that way?
Response 9: Line 111: To explain the reason for this electrode arrangement, we have cited additional sources of literature: Sharma, P.; Panta, T.; Ugiliweneza, B.; Bert, R.J.; Gerasimenko, Y.; Forrest, G.; Harkema, S. Multi-Site Spinal Cord Transcutaneous Stimulation Facilitates Upper Limb Sensory and Motor Recovery in Severe Cervical Spinal Cord Injury: A Case Study. J. Clin. Med. 2023, 12, 4416. https://doi.org/10.3390/jcm12134416. Line 110.
Comments 10: Line 131: when was 5 Hz and when 30 Hz? As they cannot be both at the same time.
Response 10: Thank you for pointing this out to us. The 5 Hz or 30 Hz stimulation was applied on different days.
The stimulation frequency was 5Hz or 30Hz.
Comments 11: Figure 6: explanations of the meaning of “i”, “S”, “trapz” are missing. Unless authors provide further explanations, it is not clear the utility of the figure.
Response 11: Your comment led us to rethink the necessity of this figure. We decided to limit ourselves to a verbal description of the definition of 60% of the power of the stabilogram spectrum.
Lines 167-173: Amplitude spectra of the fast Fourier transform with a Hamming window were found for signals from the force platform via a script developed in MATLAB R2019a [33] (Figure 5). The frequency was plotted on the abscissa axis, and the region of interest was defined as 0.02–5 Hz. The signal amplitude was plotted on the vertical (ordinate) axis. The frequency corresponding to 60% of the power was identified at the point corresponding to 60% of the signal area across the entire frequency range of 0.02–5 Hz. This value represented 60% of the power of the stabilogram spectrum, which was designated as 60%Pw.
Comments 12: Line 201: indicate version of the software and manufacturer.
Response 12: The MedStat and MedCalc v.15.1 software package (MedCalc Software bvba, Ostend, Belgium) is designed for use in medical and biological research. Line 244-245
Comments 13: Line 205: groups or interventions? It seems like there are no groups, the same participants for every intervention.
Response 13: Your assertion is indeed correct; the experiment involved a total of 14 participants.
Lines 250-254: A nonparametric approach, specifically the Wilcoxon test, was employed for the purpose of comparison between the various experimental conditions.
The significance of the differences between the parameters during stimulation with varying strengths and frequencies was calculated via the Mann–Whitney U test.
Comments 14: Figure 13: Check if authors intended to mean “surfase” or “surface”. Please correct.
Response 14: Soft surface, of course
Comments 15: Line 324: in what way authors considered that postural intability benefit from tSCS? Regarding participants are healthy people, what condition is promoting the tSCS?
Response 15:
Changes in the center of pressure position parameters and frequency response spectra demonstrate postural stability during tSCS. Our results showed that tSCS altered the feedback of postural control, resulting in slower (lower mean CoP displacement rate, ALV), more stable (lower mean CoP standard deviation, Q) and more regulated (lower CoP deviation, especially in the frontal plane, and reduced ellipse) CoP oscillations in almost all the samples.
Although sensorimotor interactions in the control of upright posture have been extensively studied, more research is still needed to improve our understanding of the mechanisms of human postural control.
To explain the mechanisms of postural stability, studies have been conducted with one of the sensory inputs switched off (e.g., eye closure or an unstable support surface). Such switching contributes to a better understanding of the role of this sensory information in maintaining upright posture. The application of additional stimuli of afferent information of a different modality, such as noninvasive spinal cord stimulation or virtual reality technologies, may expand the understanding of the mechanisms of sensorimotor interaction. The advent of noninvasive spinal cord stimulation techniques allows us to investigate spinal and brainstem mechanisms for maintaining upright posture in healthy humans. It is possible that such methods could help maintain and improve postural stability in elderly individuals or optimize training in athletes.
We have made changes to the text of the manuscript. Lines 419-424
Comments 16: Line 454: I suggest authors to consider follow-up of the results as a limitation, anyway it is just a suggestion.
Response 16: We agree with the reviewer and have expanded on our position in the Limitations section.
Lines 568-578: We have gathered extensive experimental material that cannot be adequately presented in a single article due to its complexity and volume. This experience has been our first step in discussing the results obtained. However, we are confident that the results have opened up significant opportunities for further analysis and interpretation, which require more in-depth and detailed research.

Reviewer 3 Report
Comments and Suggestions for Authors
The Authors analyzed the effect of tSCS (transcutaneous electrical stimulation) applied with different frequencies and "forces" on the mechanisms of maintaining postural stability in healthy participants.
They used the stabilographic test to verify the effects under the conditions (1) no stimulation – 1 minute; (2) 3 minutes – 5 Hz / 30 Hz stimulation with subthreshold/superthreshold stimulus strength, (3) 1 minute – without stimulation after tSCS. The change in stabilographic indices was equivocal during tSCS, improvement in postural control was obtained at 30 Hz tSCS with suprathreshold force. The spectral analysis referred to the influence of the vestibular and muscular components.The Authors concluded that transcutaneous electrical stimulation of the cervical spinal cord is an effective way to activate neural spinal locomotor networks able to modulate postural control systems.
The paper scientifically sounds. However, some minor flaws or improvements should be corrected or introduced as follows:
1. The Authors provided a very detailed and interesting study on transcutaneous electrical stimulation of the cervical spinal cord (tSCS) aiming, as the Authors suggested (lines 14,15), to improve postural stability in humans. There were 50 healthy volunteers of both sexes in this study, so the question arises: why did the Authors aim to improve the postural stability if the subjects postural abilities were not disturbed, and they were healthy? The Authors changed the experimental conditions of standing as follows EOHS – standing on a hard surface with open eyes, ECHS – standing on a hard surface with closed eyes, EOSS – standing on a soft surface with open eyes, ECSS – standing on a soft surface with closed eyes; I would recommend to consider changing the title of the study like “Different Factors influencing the postural stability during transcutaneous electrical stimulation of the cervical spinal cord".
2. “tSCS force” should be replaced with “tSCS strength” in Abstract
3. The sentence in lines 32-33 sounds sophisticated and somehow unclear - should be re-written
4. The issue "The method of transcutaneous electrical stimulation of the spinal cord (tSCS) is used to study the function of balance, motor function" (lines 44-45) should be more developed with appropriate references.
5. Replace “lumbosacral thickening” with ”lumbosacral neuromeres”
6. More demographic and anthropometric data should be provided on the study group in 2.1. Subjects of the study
7. More data on the parameters of neurophysiological recordings aiming the tSCS electrode positioning should be provided.
8. Provide the Abbreviations for the shorts used in Figure 6.
9. How was the sample size evaluated necessary to obtain the reliable results? 2.6. Statistical analysis
10. Discussion - The influence of the vestibular and muscular components in the postural coordination based on the previous reports and the current study could be better discussed.
11. Please develop in the Discussion section the context included in your Conclusions: …” Our study emphasizes the need for further research to develop effective rehabilitation techniques and improve postural stability in patients with movement disorders.”… . How do you imagine using the study’s results in rehabilitation?
Comments on the Quality of English LanguageEnglish is sometimes difficult to understand.
Author Response
- Summary
Thank you very much for taking the time to review this manuscript. We appreciate the reviewer's comments regarding the importance of this study. The authors are grateful to the Reviewer for his careful reading and comments. Please find the detailed responses below and the corresponding revisions.
Comments 1: The Authors provided a very detailed and interesting study on transcutaneous electrical stimulation of the cervical spinal cord (tSCS) aiming, as the Authors suggested (lines 14,15), to improve postural stability in humans. There were 50 healthy volunteers of both sexes in this study, so the question arises: why did the Authors aim to improve the postural stability if the subjects’ postural abilities were not disturbed, and they were healthy? The Authors changed the experimental conditions of standing as follows EOHS – standing on a hard surface with open eyes, ECHS – standing on a hard surface with closed eyes, EOSS – standing on a soft surface with open eyes, ECSS – standing on a soft surface with closed eyes; I would recommend to consider changing the title of the study like “Different Factors influencing the postural stability during transcutaneous electrical stimulation of the cervical spinal cord".
Response 2: We agree with the reviewer and have decided to change the title of the article to "Different Factors influencing the postural stability during transcutaneous electrical stimulation of the cervical spinal cord".
Comments 2: “tSCS force” should be replaced with “tSCS strength” in Abstract
Response 2: We apologize to the reviewer for the insufficient quality of the text and thank you for your comments to improve its quality. We have corrected the text: Line 19.
Comments 3: The sentence in lines 32-33 sounds sophisticated and somehow unclear - should be re-written
Response 3: Thank you for pointing this out. We have split this sentence into two sentences. We replaced the word "complicated" with the word "complex" because it was a mistake.
Lines 32-35: Balance control is a complex process that involves the voluntary and involuntary maintenance of an upright posture, both in the absence of movement and during movement. This also requires the ability to respond effectively to external and internal signals that can destabilize posture [1].
Comments 4: The issue "The method of transcutaneous electrical stimulation of the spinal cord (tSCS) is used to study the function of balance, motor function" (lines 44-45) should be more developed with appropriate references.
Response 4: Thank you. The information was edited, and relevant links were added.
Lines 56-66: The tSCS method improves motor function in neurological patients [8, 10-12], facilitates pacing movements in healthy subjects [9], and contributes to the improvement of balance function [Sayenko, D. G., Rath, M., Ferguson, A. R., Burdick, J. W., Havton, L. A., Edgerton, V. R., & Gerasimenko, Y. P. (2019). Self-Assisted Standing Enabled by Non-Invasive Spinal Stimulation after Spinal Cord Injury. Journal of neurotrauma, 36(9), 1435-1450. https://doi.org/10.1089/neu.2018.5956; Roberts, B. W. R., Atkinson, D. A., Manson, G. A., Markley, R., Kaldis, T., Britz, G. W., Horner, P. J., Vette, A. H., & Sayenko, D. G. (2021). Transcutaneous spinal cord stimulation improves postural stability in individuals with multiple sclerosis Multiple sclerosis and related disorders, 52, 103009. https://doi.org/10.1016/j.msard.2021.103009]. tSCS, like epidural stimulation, is capable of altering excitability through the recruitment of posterior-root fibers to activate interneuronal networks, potentiate the generation of postsynaptic excitatory potentials and contribute to the excitation threshold of interneurons and motor neurons. This allows the spinal neuronal network to respond to the descending impulse and increases the overall excitability of the spinal network and potentially the motor cortex [Hofstoetter, U. S., Freundl, B., Binder, H., & Minassian, K. (2018). Common neural structures activated by epidural and transcutaneous lumbar spinal cord stimulation: Elicitation of posterior root–muscle reflexes. PloS one, 13(1), e0192013. https://doi.org/10.1371/journal.pone.0192013; Benavides, F. D., Jo, H. J., Lundell, H., Edgerton, V. R., Gerasimenko, Y., & Perez, M. A. (2020). Cortical and Subcortical Effects of Transcutaneous Spinal Cord Stimulation in Humans with Tetraplegia The Journal of neuroscience : the official journal of the Society for Neuroscience, 40(13), 2633-2643. https://doi.org/10.1523/JNEUROSCI.2374-19.2020 ]. However, unlike epidural stimulation, CESSM does not require surgical implantation of electrodes. A special configuration of pulses with carrier frequencies from 5 to 10 kHz minimizes discomfort when using the CESSM [Singh G, Lucas K, Keller A, Martin R, Behrman A, Vissarionov S, Gerasimenko YP. Transcutaneous Spinal Stimulation From Adults to Children: A Review. Top Spinal Cord Inj Rehabil. 2023 Winter;29(1):16-32. doi: 10.46292/sci21-00084. ; Gorodnichev, R. M., Pivovarova, E. A., Pukhov, A., Moiseev, S. A., Savokhin, A. A., Moshonkina, T. R., Shcherbakova, N. A., Kilimnik, V. A., Selionov, V. A., Kozlovskaia, I. B., Edgerton, V. R., & Gerasimenko, I. uP. (2012). Fiziologiia cheloveka, 38(2), 46-56; Gerasimenko, Y., Gorodnichev, R., Moshonkina, T., Sayenko, D., Gad, P., & Reggie Edgerton, V. (2015). Transcutaneous electrical spinal-cord stimulation in humans. Annals of physical and rehabilitation medicine, 58(4), 225-231. https://doi.org/10.1016/j.rehab.2015.05.003].
Comments 5: Replace “lumbosacral thickening” with ”lumbosacral neuromeres”
Response 5: Line 67. We referred to Latin intumescentia lumbosacralis = English lumbar-sacral enlargement [Grossman, R. G., Tang, X., & Horner, P. J. (2022). Stereotaxic atlas of the human lumbar–sacral spinal cord. World neurosurgery, 166, e460-e468. https://doi.org/10.1016/j.wneu.2022.07.017]. Made changes.
Comments 6: More demographic and anthropometric data should be provided on the study group in 2.1. Subjects of the study
Response 6: All of our participants were volunteers. For clarification, we have included more information about them and the selection criteria.
Lines 93-99. Fourteen volunteers (3 males, 11 females, aged 18-27 years) were included in the study. The height of the participants was 163.71 ± 10.75 cm, and their weight was 59.29 ± 16.96 kg. All the participants rated themselves as healthy on the day of the study. The specific inclusion criteria were the absence of implanted metallic or electrical devices, pregnancy, and musculoskeletal or neurological disorders that could affect postural balance and well-being on the day of the study. The anthropological data of all the participants are also presented in Table S1 in the supplementary material.
Table S1. Anthropometric data of the participants
|
№ |
Participants |
Sex |
Height, cm |
Weight, kg |
Age, years |
|
1 |
p1 (BL) |
female |
157 |
50 |
25 |
|
2 |
p2 (PA) |
female |
159 |
57 |
20 |
|
3 |
p3 (MiA) |
female |
156 |
48 |
19 |
|
4 |
p4 (KS) |
female |
158 |
55 |
19 |
|
5 |
p5 (DS) |
female |
158 |
46 |
19 |
|
6 |
p6 (SA) |
female |
161 |
51 |
18 |
|
7 |
p7 (АI) |
male |
178 |
100 |
20 |
|
8 |
p8 (TR) |
male |
185 |
70 |
24 |
|
9 |
p9 (OV) |
male |
186 |
90 |
19 |
|
10 |
p10 (ShL) |
female |
157 |
65 |
19 |
|
11 |
p11 (ShV) |
female |
159 |
45 |
27 |
|
12 |
p12 (ZhA) |
female |
159 |
46 |
25 |
|
13 |
p13 (PA2) |
female |
156 |
48 |
22 |
|
14 |
p14 (NM) |
female |
163 |
59 |
21 |
|
Mean |
163.71 |
59,29 |
21,21 |
||
|
Standard dev. |
10.75 |
16,96 |
2,88 |
||
Comments 7: More data on the parameters of neurophysiological recordings aiming the tSCS electrode positioning should be provided.
Response 7: The description of the transcutaneous spinal cord stimulation procedure has been modified and links to sources describing the method have been added:
Lines 105-125: In this study, a five-channel BIOSTIM-5 stimulator (Cosyma Ltd., Moscow, Russia) was employed. A stimulating cutaneous round electrode (cathode) 32 mm in diameter with an adhesive layer (PG479/32, Fiab, UK) was positioned on the skin between the spi-nous processes of the C5 and C6 vertebrae. Rectangular electrodes (anodes) 45 × 80 mm in size with an adhesive layer (PG472W, Fiab, UK) were placed symmetrically on the clavi-cles (Figure 1A) [Sharma, P.; Panta, T.; Ugiliweneza, B.; Bert, R.J.; Gerasimenko, Y.; Forrest, G.; Harkema, S. Multi-Site Spinal Cord Transcutaneous Stimulation Facilitates Upper Limb Sensory and Motor Recovery in Severe Cervical Spinal Cord Injury: A Case Study. J. Clin. Med. 2023, 12, 4416. https://doi.org/10.3390/jcm12134416]. The stimulation was conducted via rectangular bipolar pulses with a duration of 1 ms, filled with a carrier frequency of 10 kHz, as previously described [30]. The current intensity was selected for each participant by gradually increasing the current until a motor response was observed in all the muscles of the upper limbs that were being studied: the m. flexor carpi ulnaris, the m. extensor carpi radialis, bilaterally, without allowing the appearance of any unpleasant sensations. The current intensity was subsequently reduced by approximately 10% (subthreshold stimulation) or increased by approximately 50% (threshold stimulation). The current intensity varied from 12 to 27 mA. The stimulation frequency was set at 5 Hz or 30 Hz, depending on the experimental conditions.
Comments 8: Provide the Abbreviations for the shorts used in Figure 6.
Response 8: Your comment led us to rethink the necessity of this figure. It was decided to limit ourselves to a verbal description of the definition of 60% of the power of the stabilogram spectrum:
Lines 167-177: Amplitude spectra of the fast Fourier transform with a Hamming window were found for signals from the force platform via a script developed in MATLAB R2019a [33] (Figure 5). The frequency was plotted on the abscissa axis, and the region of interest was defined as 0.02–5 Hz. The signal amplitude was plotted on the vertical (ordinate) axis. The frequency corresponding to 60% of the power was identified at the point corresponding to 60% of the signal area across the entire frequency range of 0.02–5 Hz. This value represented 60% of the power of the stabilogram spectrum, which was designated as 60%Pw.
Comments 9: How was the sample size evaluated necessary to obtain the reliable results? 2.6. Statistical analysis
Response 9: It is difficult to judge which internal mechanism of postural stability may be primary for a particular participant. In addition, we had no hypothesis about possible changes after exposure to various factors. Therefore, a preliminary estimate of the sample size was not made.
A typical sample size for studies of postural stability or transcutaneous spinal cord stimulation is 10 to 20 participants.
Comments 10: Discussion - The influence of the vestibular and muscular components in the postural coordination based on the previous reports and the current study could be better discussed.
Response 10: We agree with the reviewer .The discussion has been updated with new information regarding the vestibular and proprioceptive influences:
Lines 512-520: In addition, during TSCS, we observed a higher percentage of the CoP signal in the low-frequency (0.2–1 Hz) range and a lower percentage in the high-frequency (2–5 Hz) range. It is known that the low and moderate frequency ranges can be associated with the vestibular and muscular proprioceptive systems, respectively [60]. Activation of vestibular reflexes is critically important, especially with closed eyes and on a soft surface, since visual and proprioceptive signals are absent, and there is additional activation of the cervical spine [61]. Our studies have shown that the vestibular system is most active under such conditions. The vestibular system is believed to function as an internal cue, possibly suppressing misleading additional cues when present [62].
Lines 551-554: It has been suggested that higher frequency cervical tSCS may provide improved voluntary control during stance [72]. Another mechanism may be activation of cortical networks by cervical tSCS facilitating corticospinal descending control [19].
Comments 11: Please develop in the Discussion section the context included in your Conclusions: …” Our study emphasizes the need for further research to develop effective rehabilitation techniques and improve postural stability in patients with movement disorders.”… . How do you imagine using the study’s results in rehabilitation?
Response 11: Thank you for stopping at this point. We would like to express our gratitude for your attention. Please be advised that amendments have been made to the text.
Our analyses of changes in the contribution of sensory information during the maintenance of postural stability add to the knowledge of the mechanisms of regulation of the postural stability in the absence or alteration of information from one or more afferent inputs.
Lines 568-577: Our observations help to elucidate the mechanisms contributing to the use of tSCS for further research in evaluating the efficacy and selection of optimal parameters of electrical stimulation to improve postural control in humans to improve motor function after neurological injury. It is possible that such methods could help maintain and improve postural stability in elderly individuals or optimize training in athletes.

Round 2
Reviewer 1 Report
Comments and Suggestions for Authors
This manuscript revision suggests a major effort on the part of the authors. However, correction of remaining word errors and clarification in both the methods and results sections is needed. See specific comments in the pdf.

several minor corrections are necessary
Author Response
|
Кesponse to Reviewer 1 Comments 1. Summary Dear Reviewer, Thank you for your thoughtful review of our work and for appreciating our efforts in addressing the errors. Your feedback has been invaluable in helping us enhance the quality of our project. We truly appreciate your time and support. |
Comments 1: suggest the phrase "may be" since one study's resuls with a small sample should be careful about making definitive conclusions
Response 1: Yes, we agree and have made changes – line 26.
Comments 2: suggest: "has also been found to useful by restoring the functions of the upper extremities."
Response 2: Yes, we agree and have made changes – line 70.
Comments 3: Line 71 suggest: "...appears to modulate...".
Response 3: You are correct; we have made this replacement. Thank you for pointing that out. Line 71.
Commets 4: Line 73 This sentence seems incomplete. Do you mean that tSCS to these levels appear improve walking quality? Please clarify this.
But I have a question. If the participants why do they need the quality of their walking improved?
Response 4: This sentence is complete. In this study, simultaneous stimulation of several segments of the spinal cord was applied, resulting in an increased amplitude of involuntary stepping. To answer your question about the necessity of improving walking in healthy participants, understanding the mechanisms of transcutaneous spinal cord stimulation on motor activity, as well as developing a stimulation algorithm needed for restoring movement in patients, is important.
Comments 5: Line 79 patients with what type of diagnosis and pathology?
Response 5: Thank you for the feedback. We added the following: patients with spinal cord injury.
Lines 77--79: The application of higher frequencies results in a more stable position of the center of pressure (CoP) and makes it easier for patients with SCI to stand independently [16].
Comments 6: Lines 83; need to write in the past tense, so I suggest: "...cord would be an effective..."
Response 6: Thank you for pointing this out. Therefore, we have replaced “is” with “cord would be effective” (lines 84--85).
Comments 7: Lines 94 since your units are kg, this should be body mass not weight
Response 7: Thank you; we agree with the feedback and have made changes.
Lines 95--96: The height of the participants was 163.7 ± 10.7 cm, and their body mass was 59.2 ± 16.9 kg.
Comments 8: Lines 124 please describe what you did after electrode placement to verify the accuracy and strength of the EMG signal.
Response 8: Our equipment allows the quality of electrode placement to be checked. On the front panel of the device, there is a button to initiate impedance measurement, enabling the researcher to verify the quality of electrode placement immediately after positioning it on the patient, without leaving the patient or needing to operate the computer. The impedance values of all electrodes are displayed by LEDs on the front panel of the neuromyograph. Additionally, the button allows monitoring and recording of the EMG signal.
We have added a note to the procedure description:
Lines 135-136. After placing the electrodes, the quality of their installation was checked by measuring the subelectrode impedance.
Comments 9: Lines 139-140 a force platform does not record a COP but the ground reaction forces which are put into a formula which calculates a center of pressure; please clarify exactly what you did?
Response 9: Thank you for pointing this out. A Stabilan-01-2 force plate system (Rhythm Ltd., Taganrog, Russia) with StabMed 2.13 software was used for stabilometry. The system recorded the CoP positions with a sampling frequency of 50 Hz and a resolution of <0.01 mm. "Stabilan" is a hardware complex designed for recording, processing, and analyzing the trajectory of the center of pressure of the human body on a support surface. The registration CoP fluctuations and all calculations were provided by the corresponding software of the StabMed 2 stability analyzer.
Comments 10: Lines 157 was the force plate zeroed or re-calibrated prior to each test trial? if not, please describe what you did instead. If you did not zero the place between each and every condition any residual forces may have influenced your force plate and COP results (a potentially significant methodological limitation)
Response 10: We respectfully agree with this comment. Yes, of course. Before testing, centering was carried out so that the position of the human pressure center coincided with the origin of the coordinates on the computer monitor via the program indicated by the red square in Figure 5. Line 216.
Comments 11: Lines 157 one sentence paragraphs are not acceptable; please find a way to include all of the information related to collection of ground reactions forces and the software that is used to determined the AP and ML COP in one paragraph
Response 11: We agree. We have combined this sentence with the previous paragraph.
Comments 12: Lines 239 Since you conducted the tests on different days, please provide evidence regarding the accuracy of all measuring devices, such as the dynamometer platform, EMG system, etc. If you do not know the accuracy of your devices, this is another potentially significant limitation of the study that should be pointed out.
Response 12: We have provided data on our devices. All devices are state-certified and meet measurement quality standards.
Comments 13: Lines 265 some bars have oblique lines...what does that mean?
Response 13: Lines 268-269: The data collected during stimulation were divided into three equal one-minute intervals (bars with lines).
Comments 14: Lines 285 when I look at figure 9 it is difficult to see an increase; with more explanation in the manuscript you can help the reader see what is going on in the figure
Response 14: We understand that it may be difficult to see in the figure, but we would like to draw your attention to the white bars and the bars with diagonal lines. Tables 3 and 4 can also be found in the supplementary material; we have added these links to the text of the manuscript.
Comments 15: Lines 285 root mean square distance of what?
Response 15: Thank you for pointing this out. Thank you for pointing this out. This is a mistake, and we have replaced it with the term "root mean square deviation". Line 286
Comments 16: Lines 286 again, what do the oblique hash marks indicate?
Response 16: Thank you for pointing this out. The data collected during stimulation were divided into three equal one-minute intervals (bars with lines).
Comments 17: Lines 348 please explain the oblique hash marks; you need to label the figures with "a" and "b".
Response 17: Thank you for pointing this out. Yes, we missed the figure captions. We have added the letters and have included the word 'slashes' in the figure captions.
«bars with slashes».
Comments 18: Lines 412 should be "was'; keep looking for, and correcting, word tense errors Response 18: We agree.
Comments 19: Lines 414 Prior to restating the purpose of this research, reiterate the problem you have identified and the rationale/need for this study
Response 19: You are correct. We added the following paragraph:
Lines 416-420. With the widespread use of spinal cord electrical stimulation in locomotion and postural control studies, much of the role of spinal networks remains unclear. Transcutaneous spinal cord stimulation (tSCS) offers a significant advantage: this noninvasive method activates spinal networks, including those in healthy individuals, enabling the study of spinal cord neural networks in humans.
Comments 20: Line 424. It looks like you said the same thing twice.
Response 20: Yes, we removed one sentence.
Comments 21: Line 531; 534 should be "participants"; please comb your entire manuscript again to make sure your are using "participants" everywhere
Response 21: Yes, we will check everything.
Comments 22: Line 567. what kind of studies, i.e., EMG?
Response 22: Additional EMG studies on ankle and hip muscle activity would be very valuable, given their importance in postural control.
